# SynBench: Evaluating Pretrained Representations for Image Classification using Synthetic Data

## Abstract

Fine-tuning large models pretrained at scale on broad data for solving downstream tasks has made considerable success in recent years. There seems to be indeed an ongoing paradigm shift in deep learning from task-centric model design to task-agnostic representation learning and task-specific fine-tuning. Specifically, the representations of pretrained models are used as a foundation for different downstream tasks. This paper proposes a new task-agnostic framework, *SynBench*, to measure the quality of pretrained representations for image classification using synthetic data. To address the challenge of task-agnostic data-free evaluation, we design synthetic binary classification proxy tasks with class-conditional Gaussian mixtures. This way we probe and compare the robustness-accuracy performance on pretrained representations and input synthetic data. SynBench offers a holistic quantitative evaluation, informs the model designers of the intrinsic performance, and spares efforts on task-specific finetuning with real-life data. Evaluated with various pretrained vision models for different downstream image classification tasks, the experimental results show that our SynBench score matches well the actual linear probing performance of the pretrained model when fine-tuned on downstream tasks using real-life data. Finally, SynBench can also be used in robust linear probing to mitigate the robustness-accuracy tradeoff in downstream tasks.

## 1 Introduction

In recent years, the use of large pretrained neural networks for efficient fine-tuning on downstream tasks has prevailed in many application domains such as vision, language, and speech. Instead of designing task-dependent neural network architectures for different downstream tasks, the current methodology focuses on the principle of task-agnostic pretraining and task-specific finetuning. This methodology uses a neural network pretrained on a large-scale broad dataset to extract generic representations of the input data, which we call *pretrained representations* for simplicity. The pretrained representations are then used as a foundation (Bommasani et al., 2021) to solve downstream tasks. Prevalent ways include training a linear head (i.e., linear probing) on the representations with the labels provided by a downstream dataset, or by simply employing zero-shot inference.

When gauging the usefulness of a pretrained model, it is a convention to conduct evaluations on selected real-life tasks. For example, ViT (Dosovitskiy et al., 2020) reports accuracy on 25 tasks, CLIP (Radford et al., 2021) probes models on 27 datasets, and PLEX (Tran et al., 2022) devises 10 types of tasks over 40 datasets to systematically evaluate different aspects of reliability on both vision and language domains. However, this convention has several drawbacks. For example, the evaluation process evidently poses significant computational overhead on the model trainer and raises data privacy concerns, setting a high bar for new model designs and large-scale AI governance. More importantly, the evaluation result is dependent on specific evaluation datasets. Thus the nominal evaluation score can be inconclusive if the evaluation data are biased or under-representative. For instance, ViT-L/16 is reportedly performing better than ViT-B/16 on 23 out of 27 linear probing tasks according to (Radford et al., 2021, Table 10), but worse than ViT-B/16 on FoodSeg103 (Wu et al., 2021, Table 8), X-ray images (Okolo et al., 2022, Table 4-8), and magnetic resonance imaging (Tummala et al., 2022, Table 2-3) tasks. In essence, a poor probing result might come from either (1) evaluation data bias or (2) true model deficiency, or both. In this paper, we establish our evaluation benchmark by disentangling the effect of the two and focusing on designing sanity checks for the latter. We utilize synthetic data generated from a class-conditional data prior, whose optimal classification

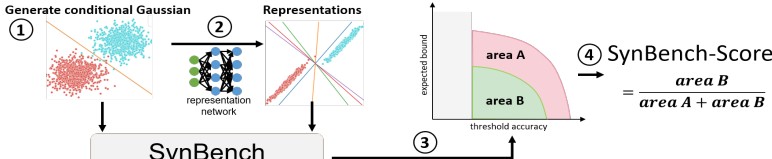

Figure 1: Overview of SynBench. Step 1: generate class-conditional Gaussian and form the inputs to the pretrained model; Step 2: gather rendered representations; Step 3: measure the expected robustness bound under a range of threshold accuracy for both input synthetic data and their representations according to **eqn. (2)** and obtain the expected bound-threshold accuracy plot; Step 4: calculate SynBench score by the relative area under the curve of the representations (area B) to the inputs (area A + area B) in the expected bound-threshold accuracy plot. The closer the ratio is to 1, the better the quality of pretrained representations is, in terms of the robustness-accuracy characterization.

strategy is known. We use them to compare with representations' linear separability. For example, Fisher's linear discriminant rule (Johnson et al., 2002; Petridis & Perantonis, 2004) decides the optimal strategy for Gaussian distribution. If the data can be separated with 90% accuracy in the raw input space and 60% accuracy in the representation space, then the pretrained model has an intrinsic deficiency. On top of that, the trending practice of pretraining and fine-tuning also signifies immediate damage to all downstream tasks if the underlying pretrained model has hidden risks, such as lacking robustness to adversarial examples. It is worth noting that these types of risks may not be informed by the standard accuracy as they do not correlate well (e.g. even negative correlation as pointed out by Su et al. (2018)). Luckily, similar to Fisher's linear discriminant rule for the optimal standard accuracy, Dan et al. (2020) has characterized the optimal classification strategy in the presence of input perturbations. Our sanity check can thereby also evaluate the adversarial robustness of pretrained models by considering the same synthetic conditional Gaussian data prior. Our use of Gaussian mixtures for analysis is supported by its capability of modeling the statistics of natural images (Zoran & Weiss, 2012) and prior arts in the topic of Gaussian design (Donoho & Tanner, 2009; CANDÈS & SUR, 2020; Bartlett et al., 2020). Besides being an universal approximator, the fact that Gaussian mixture models often lead to mathematically tractable problems (as in this paper) also give rise to a recent line of work (Mignacco et al., 2020; Refinetti et al., 2021; Loureiro et al., 2021) that analyze the asymptotic performance of a large class of machine learning problems in the proportional high-dimensional limit under the Gaussian mixture data assumption (Dandi et al., 2023). This paper also serves as an evidence of how Gaussian mixtures can take part in model evaluations.

An ideal pretrained model should entail both good accuracy and adversarial robustness, and the level of goodness is desired to be measurable in a task/data-agnostic manner. In this paper, we propose *SynBench* to precisely address this requirement. Specifically, SynBench establishes a theoretical reference characterizing the robustness-accuracy tradeoff of the synthetic data based on the Bayes optimal linear classifiers. Then, SynBench obtains the representations of the same synthetic data from the pretrained model and compares them to the reference. Finally, we define the ratio of area-under-the-curves in robustness-accuracy plots as a quantifiable metric of the pretrained representation quality. The entire procedure of SynBench is illustrated in Figure 1. We list possible use case of SynBench in the Appendix Section A.1. SynBench features the following key advantages:

1. *Soundness*: We formalize the fundamental tradeoff in robustness and accuracy of the considered conditional Gaussian model and use this characterization as a reference to benchmark the quality of pretrained representations in a completely real-data-free scenario.

2. *Task-independence*: Since the pretraining of large models is independent of the downstream datasets and tasks (e.g., through self-supervised or unsupervised training on broad data at scale), the use of synthetic data in SynBench provides a task-agnostic approach to evaluating pretrained representations without the knowledge of downstream tasks and datasets.

3. *Completeness and privacy*: The flexibility of generating synthetic data (e.g., by adopting a different data sampling procedure) offers a good proxy towards a more comprehensive evaluation of pretrained representations prior to fine-tuned on different downstream datasets, especially in the scenario when the available datasets are not representative of the entire downstream datasets. Moreover, the use of synthetic data enables full control and simulation over data size and distribution, protects data privacy, and can facilitate model auditing and governance.

We highlight our **main contributions** as follows:

- We propose SynBench, a novel task-agnostic framework that uses data synthesized from a data prior to evaluate the quality of pretrained representations. The evaluation process is independent of the downstream image classification datasets/tasks.

- Evaluated with several pretrained vision models for image classification, our experimental results show that the metric provided by SynBench matches well the model performance in terms of adversarial robustness and standard accuracy when finetuned on several downstream datasets. For example, SynBench-Score suggests that the Imagenet21k pretrained network (*ViT-B/16-in21k*) improves with finetuning on Imagenet1k (*ViT-B/16*), echoing with the higher linear probing accuracy of *ViT-B/16* on real-life datasets. The Pearson correlation coefficient between SynBench-Scores and the average real-life task accuracy is larger than $0.9$.

- We show that SynBench can be used to inform the design and selection of the hyperparameters in robust linear probing to mitigate the robustness-accuracy tradeoff when fine-tuned on downstream datasets. For example, conducting $\epsilon$-robust linear probing with $\epsilon$ selected by SynBench-Score gives *ViT-B/16* $0.1\%$ and $2.7\%$ increase in CIFAR10 standard and robust accuracy and $0.7\%$ and $2.5\%$ increase in TinyImagenet standard and robust accuracy.

## 2 RELATED WORK

**Pretrained models in vision.** In the past few years, much focus in the machine learning community has been shifted to training representation networks capable of extracting features for a variety of downstream tasks with minimal fine-tuning. Nowadays, many common vision tasks are achieved with the assistance of good backbones, e.g. classifications (Yu et al., 2022; Wortsman et al., 2022; Foret et al., 2020; Xie et al., 2020; Dosovitskiy et al., 2020; Chen et al., 2020a), object detection (Redmon & Farhadi, 2017; Liu et al., 2016), segmentation (Chen et al., 2017; Xie et al., 2021), etc. Among the popular backbones, vision transformers (ViT) (Dosovitskiy et al., 2020) and convolutional models (e.g. ResNet He et al. (2016)) have attracted enormous interest. We will exemplify the use of SynBench using several pretrained ViTs and ResNets.

**Benchmarking pretrained models.** Since pretrained models are used as a foundation for different downstream tasks, it is central to transfer learning (Neyshabur et al., 2020; Pruksachatkun et al., 2020), and also tightly related to model generalization (Qiao et al., 2020; Carlucci et al., 2019). To benchmark the performance of a pretrained model, it is a convention to apply the pretrained model for a number of popular tasks and conduct linear probing on the representations (Chen et al., 2020b; Dosovitskiy et al., 2020; Chen et al., 2020a; 2021). Besides accuracy-based probing methods, evaluation methods have been proposed based on information theory and minimum description length (Blier & Ollivier, 2018; Voita & Titov, 2020), surplus description length (Whitney et al., 2020), maximum evidence (You et al., 2021), Fisher discriminant analysis (Shao et al., 2022), among others. These metrics are reliant on the label information of the downstream tasks and are hence task-specific.

Lately, more fundamental questions related to pretrained models are brought up (Bommasani et al., 2021; Tran et al., 2022; Zhang & Ré, 2022; Shi et al., 2022). Bommasani et al. (2021) raised practical concerns about the homogenization incentivized by the scale of the pretraining. Although homogenization might help in achieving competitive performance for some downstream tasks, the defects are also inherited by all these downstreams. On that account, a more careful study of the fundamentals of pretrained models is of paramount importance. Tran et al. (2022) explored the reliability of pretrained models by devising 10 types of tasks on 40 datasets. It is further pointed out by Zhang & Ré (2022) in 9 benchmarks that pretrained models may not be robust to subpopulation or group shift. The adversarial robustness is benchmarked by Shao et al. (2021); Paul & Chen (2022).

**Optimal representations.** In the seminal work of deep representation theory, Achille & Soatto (2018) depicted the desired optimal representations in supervised learning to be sufficient for the downstream task, invariant to the effect of nuisances, maximally disentangled, and have minimal mutual information between representations and inputs. Focusing more on generalization than compression, Dubois et al. (2020) provided the optimal representation based on $\mathcal{V}$-information (Xu et al., 2019). Ruan et al. (2021) defined the optimal representations for domain generalization. Dubois et al. (2022) characterized idealized representations in self-supervised learning as ones that are well-distinguished by the desired family of probes for potential invariant tasks, have sufficiently large dimensions, and be invariant to input augmentations.

**Why SynBench?** To enable quantifying representation quality in the pretraining stage, SynBench differs from the above quantifiers/frameworks as it does not need knowledge of any real-world downstream data. Moreover, SynBench has full control of the evaluation set via synthetic data generation. With the assumed synthetic data distribution, we can theoretically characterize the reference robustness-accuracy tradeoff. Therefore, SynBench provides a standardized quality metric with theoretical groundings and evaluates for representations induced by pretrained models.

## 3 SynBench: Methodology and Evaluation

Without the knowledge of the downstream tasks and data, we aim to develop a task-agnostic framework to evaluate some fundamental behaviors of the representation network. In this paper, we inspect and quantify how representation networks preserve the robustness and accuracy enjoyed by the original synthesized data. On the whole, we measure the idealized robustness-accuracy tradeoff using synthetic data. By propagating the Gaussian realizations through different representation networks, we can also compare the robustness-accuracy tradeoff for representations. We start this section by giving the preliminaries on the synthetic data of interest.

### 3.1 Synthetic Data

We consider binary classification problems with data pair $(x, y)$ generated from the mixture of two Gaussian distributions $P_{\mu_1, \mu_2, \Sigma}$, such that $x|y = 1 \sim \mathcal{N}(\mu_1, \Sigma), x|y = -1 \sim \mathcal{N}(\mu_2, \Sigma)$, or equivalently,

$$x - \frac{\mu_1 + \mu_2}{2}|y = 1 \sim \mathcal{N}(\tilde{\mu}, \Sigma); \; x - \frac{\mu_1 + \mu_2}{2}|y = -1 \sim \mathcal{N}(-\tilde{\mu}, \Sigma), \tag{1}$$

where $y \in \mathcal{C} = \{+1, -1\}$, $P(y = +1) = \tau$, $P(y = -1) = 1 - \tau$, and $\tilde{\mu} = \frac{\mu_1 - \mu_2}{2}$. We focus on the class-balanced case ($\tau = \frac{1}{2}$) and defer the imbalanced case to Appendix A.5. When sampling from this idealized distribution, we eliminate the factor of data bias and can benchmark the accuracy and robustness degradation in an ideal setting.

Let $\| \cdot \|_p$ denote the $\ell_p$ norm of a vector for any $p \geq 1$. For a given classifier $f$ and input $x$ with $f(x) = y$, where $y$ is the predicted label, it is not rational for the classifier to respond differently to $x + \delta$ than to $x$ for a small perturbation level measured by $\|\delta\|_p$, i.e. inconsistent top-1 prediction (Szegedy et al., 2013; Goodfellow et al., 2014). Therefore, the level of (adversarial) robustness for a classifier can be measured by the minimum magnitude of perturbation that causes misclassification, i.e. $\|\Delta\|_p := \min_{\delta: f(x+\delta) \neq f(x)} \|\delta\|_p$. For a generic function $f$, solving the optimization problem exactly is hard (Katz et al., 2017; Sinha et al., 2018). Luckily, one can readily solve for the optimization if $f$ is affine (Moosavi-Dezfooli et al., 2016).

### 3.2 Main Theorem

In what follows, we will leverage this point and focus on the linear classifier that minimizes robust classification error. An ideal candidate classifier for the class conditional Gaussian (equation 1) is specified by the robust Bayes optimal classifier (Bhagoji et al., 2019; Dobriban et al., 2020). Specifically, it is stated that the optimal robust classifier (with a robust margin $\epsilon$) for data generated from equation 1 is a linear classifier. We derive the following result as a direct application of the fact. To simplify the exposition, we focus on the $\ell_2$ norm in the remainder of this paper. We refer the readers to Appendix A.4 for general $\ell_p$-norm results. We use "bound" to denote the minimal perturbation of a sample. We first formally state our theorem (proofs in Appendix A.3) that serves as the foundation of our SynBench framework.

**Theorem 1.** *For any sample $x$, the optimal robust classifier $f_\epsilon$ for $P_{\mu_1, \mu_2, \Sigma}$ gives*

*(i) the bound (decision margin)*
$$\|\Delta\|_2 = \frac{|(x - \frac{\mu_1 + \mu_2}{2})^T \Sigma^{-1}(\tilde{\mu} - z_\Sigma(\tilde{\mu}))|}{\|\Sigma^{-1}(\tilde{\mu} - z_\Sigma(\tilde{\mu}))\|_2},$$

*(ii) the scaled bound* $\|\bar{\Delta}\|_2 = \frac{|(x - \frac{\mu_1 + \mu_2}{2})^T \Sigma^{-1}(\tilde{\mu} - z_\Sigma(\tilde{\mu}))|}{|\tilde{\mu}^T \Sigma^{-1}(\tilde{\mu} - z_\Sigma(\tilde{\mu}))|}.$

*For a sample $x \sim P_{\mu_1, \mu_2, \Sigma}$, it further gives*

*(iii) the standard accuracy $a = \Phi(\frac{\tilde{\mu}^T \Sigma^{-1}(\tilde{\mu} - z_\Sigma(\tilde{\mu}))}{\|\Sigma^{-1}(\tilde{\mu} - z_\Sigma(\tilde{\mu}))\|_\Sigma})$,*

*(iv) the expected scaled bound of correct samples*
$$\mathbb{E}\left[\|\bar{\Delta}\|_2 \mid f_\epsilon(x) = y\right] = \frac{1}{\sqrt{2\pi}} \frac{1}{a \Phi^{-1}(a)} e^{-\frac{1}{2}(\Phi^{-1}(a))^2} + 1,$$

*where $z_\Sigma$ is the solution of the convex problem $\arg\min_{\|z\|_2 \leq \epsilon} (\tilde{\mu} - z)^T \Sigma^{-1}(\tilde{\mu} - z)$ and $\Phi$ denotes the CDF of the standard normal distribution.*

We note that for samples drawn from $P_{\mu_1, \mu_2, \Sigma}$, $\Sigma = \sigma^2 I_d$, all $\epsilon$-robust Bayes optimal classifier overlap with each other. For a general covariance $\Sigma$, the $\epsilon$ of an $\epsilon$-robust Bayes classifier specifies the desired size of margin and demonstrates the robustness accuracy tradeoff. We give an illustrative 2D

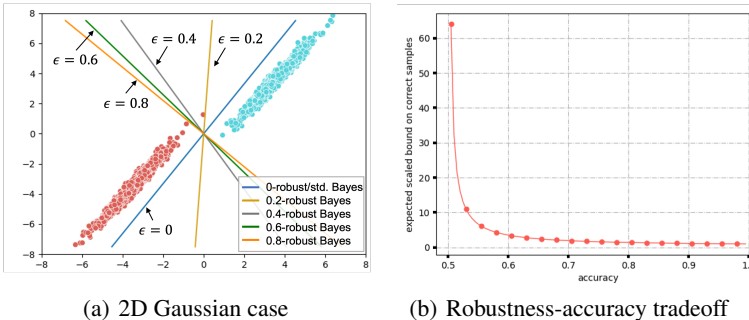

(a) 2D Gaussian case         (b) Robustness-accuracy tradeoff

Figure 2: Illustration of robustness-accuracy tradeoff suggested by $\epsilon$-robust Bayes optimal classifiers. Figure (a) depicts a class-conditional 2D Gaussian case with decision boundaries drawn by $\epsilon$-robust Bayes optimal classifiers of varying $\epsilon$ values. Figure (b) draws the theoretically characterized robustness-accuracy tradeoff given in Theorem 1(iv).

class-conditional Gaussian example in Figure 2(a), where different $\epsilon$-robust Bayes classifiers give different overall margins at the cost of accuracy. As $\epsilon$ increases, the robust Bayes optimal classifier rotates counterclockwise, leading to increased misclassifications, but also overall enlarged margins.

### 3.3 OBJECTIVE

For a given representation network parameterized by $\theta$, we are interested in evaluating the expected bounds on synthetic data and their representations, under a thresholding accuracy $a_t$. That is, $\mathbb{E}_{\mu \sim \mathbb{P}_\mu, \Sigma \sim \mathbb{P}_\Sigma, x-\bar{\mu}|y \sim \mathcal{N}(y\mu, \Sigma)} \left[ \|\bar{\Delta}\|_2 \mid f_\epsilon(x) = y, a > a_t \right]$ for $\bar{\Delta} = \bar{\Delta}_x$ and $\bar{\Delta}_z$, where $\mathbb{P}_\mu$ and $\mathbb{P}_\Sigma$ characterize the probability density function of the synthetic data manifold of interest, $\bar{\mu}$ is a translation vector allowing non-symmetric class conditional Gaussian, and $\bar{\Delta}_x$ and $\bar{\Delta}_z$ denote the bounds on synthetic data and representations respectively. Here, without the prior of applications, we assume $\mu = s \cdot 1_d / \sqrt{d}$, where $s$ denotes a random variable that follows uniform distribution and $1_d / \sqrt{d}$ is the normalized all-ones vector. For simplicity, we let $\Sigma = I_d$. Formally, we define the accuracy-constrained expected bound $E_{\theta, \epsilon}(a_t)$ as

$$E_{\theta, \epsilon}(a_t) = \mathbb{E}_{s,x} \left[ \|\bar{\Delta}\|_2 \mid f_\epsilon(x) = y, a(s, \epsilon) > a_t \right] = \sum_i \mathbb{E}_x \left[ \|\bar{\Delta}\|_2 \mid f_\epsilon(x) = y \right] \mathbb{1}_{a(s_i, \epsilon) > a_t} p(s_i), \quad (2)$$

where $\mathbb{1}_{a(s_i, \epsilon) > a_t}$ is the indicator function specifying the $s_i, \epsilon$-dependent accuracy $a$ that surpasses the threshold accuracy $a_t$. We put the detailed derivation in Appendix A.2. In the following sections, we will illustrate how to calculate the inner expectation term $\mathbb{E}_x \left[ \|\bar{\Delta}\|_2 \mid f_\epsilon(x) = y \right]$ for both the raw data (synthetic data) and representations.

**Raw data.** For raw data synthesized from $P_{\mu_1, \mu_2, \Sigma}$ according to equation 1, the inner expectation term is given by Theorem 1(iv) $\mathbb{E} \left[ \|\bar{\Delta}_x\|_2 \mid f_\epsilon(x) = y \right] = \frac{1}{\sqrt{2\pi}} \frac{1}{a\Phi^{-1}(a)} e^{-\frac{1}{2}\left(\Phi^{-1}(a)\right)^2} + 1$, where $a$ denotes the standard accuracy. The subscript $x$ in the expected scaled bound $\mathbb{E} \left[ \|\bar{\Delta}_x\|_2 \mid f_\epsilon(x) = y \right]$ indicates the raw data space, to distinguish from the scaled bound to be derived for representations. We highlight that Theorem 1(iv) directly shows a robustness-accuracy tradeoff. We plot the expected scaled bound as a function of accuracy in Figure 2(b), which holds true when the data follow equation 1 exactly. In SynBench, we treat this theoretically-derived robustness-accuracy tradeoff as the reference, enabling a fair comparison among representations induced by different pretrained models.

**Representations.** Given a pretrained network, we gather the representations of the Gaussian realizations and quantify the bound induced by robust Bayes optimal classifier in the representation space. When deriving the robust Bayes optimal classifier, we model the representations by a general conditional Gaussian $z|y = 1 \sim \mathcal{N}(\mu_1, \Sigma), z|y = -1 \sim \mathcal{N}(\mu_2, \Sigma)$. By Theorem 1(ii), we consider the optimal robust classifier for the modeled conditional Gaussian in the representation space to calculate the scaled bound $\|\bar{\Delta}_z\|_2 = \frac{|(z - \frac{\mu_1 + \mu_2}{2})^T \Sigma^{-1}(\tilde{\mu} - z_\Sigma(\tilde{\mu}))|}{|\tilde{\mu}^T \Sigma^{-1}(\tilde{\mu} - z_\Sigma(\tilde{\mu}))|}$ for correctly-classified samples and the inner expectation is estimated empirically. It should be noted that now the Bayes optimal classifier does not necessarily coincide with the robust Bayes optimal classifier even when we synthesized the dataset with an identity matrix covariance in the input space.

### 3.4 ROBUSTNESS-ACCURACY QUANTIFICATION OF REPRESENTATIONS

Recall that we aim to calculate $E_{\theta, \epsilon}(a_t) = \sum_i \mathbb{E}_{x|y \sim \mathcal{N}(ys_i \cdot 1_d / \sqrt{d}, I_d)} \left[ \|\bar{\Delta}\|_2 \mid f_\epsilon(x) = y \right] \mathbb{1}_{a(s_i, \epsilon) > a_t} p(s_i)$ for both raw data and the representations (i.e. $\|\bar{\Delta}_x\|$ and $\|\bar{\Delta}_z\|$). We treat the expected bounds of the

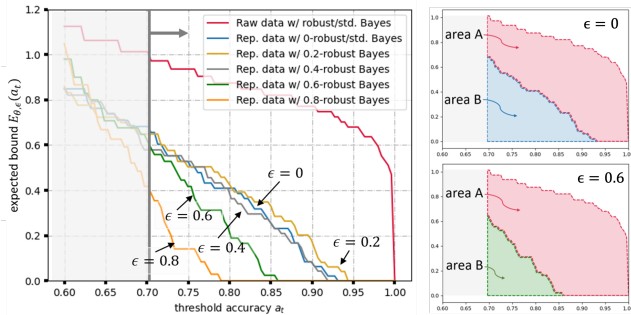

Figure 3: An example of the robustness-accuracy quantification of representations for ViT-B/16. (Left) The expected bound-threshold accuracy plot for the input raw data ($E(a_t)$) and representations ($E_{\theta,\epsilon}(a_t)$) with $\epsilon = 0 \sim 0.8$. (Right) To calculate the SynBench-Score for $\epsilon = 0$ (top) and $\epsilon = 0.6$ (bottom), we use the definition SynBench-Score$(\theta, \epsilon, a_t) = \frac{\text{area B}}{\text{area A+area B}}$ (refer to equation 3), which gives SynBench-Score$(\theta_{\text{ViT-B/16}}, 0, 0.7) = 0.33$ and SynBench-Score$(\theta_{\text{ViT-B/16}}, 0.6, 0.7) = 0.20$.

raw data under a threshold accuracy as the reference. Given a representation network, we compare the expected bounds of the representations rendered by representation networks with the reference.

In our implementation, we take $s \sim \mathcal{U}\{0.1, 5\}$ under the guidance of Theorem 1(iii). Specifically, as Theorem 1(iii) gives an analytical expected accuracy for class conditional Gaussian, we can obtain the desired range of $s$ by giving the accuracy. Since we are interested in having the reference as a class conditional Gaussian that yields accuracy from 55% to almost 100%, we set the starting and ending $s$ by the fact that $\Phi(0.1) \approx 0.55$ and $\Phi(5) \approx 1.0$. We reiterate that with more accurate modeling of the data manifold of interest, SynBench can give a more precise capture of the pretrained representation performance. We will demonstrate this point in Section 4.4.

When the data is perfect Gaussian (e.g. input synthetic data), we calculate $E_{\theta,\epsilon}(a_t)$ as detailed in Section 3.3. We note that $\bar{\Delta}_x$ is independent of pretrained network parameters $\theta$, and all the $\epsilon$-robust classifiers $f_\epsilon$ in the input space overlap with each other when $\Sigma = I_d$. We hereby denote the desired metric on the input synthetic data by $E(a_t)$, to distinguish from that on the representations $E_{\theta,\epsilon}(a_t)$. For representations, we calculate $E_{\theta,\epsilon}(a_t)$ following Section 3.3 and the expectation is estimated empirically. We show an example of the probing results in Figure 3.

To integrate over all the desired threshold accuracy, we use the area under the curve (AUC) and give the ratio to the reference by

$$\text{SynBench-Score}(\theta, \epsilon, a_T) = \frac{\int_{a_T}^{1} E_{\theta,\epsilon}(a_t) da_t}{\int_{a_T}^{1} E(a_t) da_t}, \tag{3}$$

which correspond to the relative area $\frac{\text{area B}}{\text{area A + area B}}$ in Figure 3. Values of SynBench-Score closer to 1 imply better probing performance on pretrained representations. To summarize, SynBench framework generates a sequence of proxy tasks with different difficulty levels (monitored by $s$). With each proxy task, we can obtain an accuracy and an expected bound (Section 3.3). With gathered pairs of accuracy and expected bound, we filter ones whose accuracy is below a threshold accuracy (x-axis), and calculate the accuracy-constrained expected bound to reflect the robustness level (y-axis). With this, the AUC will counter for the discriminative power of the foundation model given an idealized distribution, as well as the robustness level. We refer readers to Appendix A.6 for the pseudo-code.

## 4 EXPERIMENTAL RESULTS

In Section 4.1, we give the setup of our experiments. We exemplify the use of SynBench in making efficient comparisons of pretrained representations in Section 4.2. We compare SynBench with baseline methods and demonstrate the supremacy of SynBench-Score in giving consistent model suggestions and high correlation with performance on possible downstream tasks. In Section 4.3, we study how SynBench can be used to select robust linear probing hyper-parameters. In Section 4.4, we show how to model the covariance matrix $\Sigma$ used for synthesizing Gaussian samples given prior knowledge of the downstream data distribution.

### 4.1 EXPERIMENT SETUP AND BASELINES

In the following sections, we will calculate SynBench-Scores for pretrained models and make pairwise comparisons. For example, ViT-B/16 is a fine-tuned pretrained model from ViT-B/16-in21k. By checking their SynBench-Scores, we could understand how the fine-tuning procedure helps or

worsens the performance. In order to systematically understand how each network attribute affects the robustness-accuracy performance, it is desirable to control the variates. We list and compare 10 pretrained vision transformers (ViTs) (Dosovitskiy et al., 2020; Chen et al., 2021; Caron et al., 2021) and ResNets (Chen et al., 2020c) in Appendix Table 5.

Although to the best of our knowledge, there is no real-data-free evaluation method for pretrained representations, we refer to recent work (Whitney et al., 2020; You et al., 2021; Shao et al., 2022) and report the validation accuracy (Val loss), minimum description length (MDL), surplus description length (SDL), logarithm of maximum evidence (LogME) and self-challenging Fisher discriminant analysis (SFDA), following the official implementation from the literature on our synthetic proxy task as baselines (Whitney et al., 2020; Shao et al., 2022). In essence, we expect these real-data-free evaluations for pretrained models can give meaningful performance assessments of possible downstream tasks. For this purpose, we take an average of the accuracy in 27 downstream tasks (cf. Radford et al. (2021), Table 10) as in the literature (Dosovitskiy et al., 2020; Radford et al., 2021; Li et al., 2022; Fang et al., 2023; Yu et al., 2022) to give a sense of the general performance on possible downstream tasks, and report the Pearson correlation coefficients with SynBench-Scores. Building on top of these, we also show the consistency of SynBench suggestions given different numbers of synthetic realizations compared to the baselines.

To provide a comprehensive evaluation, we give SynBench-Score$(\theta, \epsilon, a_t)$ with $a_t$ ranging from 0.7 to 0.9, and $\epsilon$ from 0 to 0.8. Due to the space limit, $a_t \neq 0.7$ and some $\epsilon$ results are deferred to the appendix. The runtime of SynBench depends on the number of outcomes of the discrete uniform distribution $\mathcal{U}\{0.1, 5\}$. For one outcome, it costs $59$ seconds to generate $2048$ Gaussian samples, $37$ and $81$ seconds to obtain the SynBench-Score for ViT-B/16 and ViT-L/16 on single GeForce RTX 2080 super. We refer the readers to Appendix A.8 for the detailed runtime analysis. Besides the SynBench-Score, we will also report the standard accuracy (SA) and robust accuracy (RA, accuracy against adversarial perturbations) for studying robustness-accuracy performance.

## 4.2 SynBench Benchmarking of Pretrained Representations

**Comparing model attributes.** We list the SynBench-Score of the 10 pretrained representations with their standard and robust accuracy on the class-conditional Gaussian proxy task in Table 1. The robust accuracy is obtained by $\ell_2$ PGD attack (Madry et al., 2018) with attack strength $0.2$.

By referring to rows "ViT-B/16" and "ViT-B/16-in21k", we see that SynBench will suggest ViT-B/16 over ViT-B/16-in21k, implying that the fine-tuning is beneficial on ViT-B/16-in21k - both networks are pretrained on Imagenet 21k with supervision, whereas ViT-B/16 is further finetuned on Imagenet 1k. We can also use SynBench to evaluate the effect of model sizes. Specifically, we refer to rows "ViT-Ti/16", "ViT-B/16", "ViT-L/16", and see that ViT-B/16 and ViT-L/16 score much higher than ViT-Ti/16, suggesting larger models have better capacities for robustness and accuracy. It is noticeable that ViT-B/16 is generally on par with ViT-L/16

Table 1: The SynBench-Score of pretrained representations and the standard/robust accuracy (SA/RA) (%) of their linear probing classifier on class-conditional Gaussian data.

| Models | SynBench-Score ($\epsilon = 0$) | SA | RA |
|---|---|---|---|
| ViT-Ti/16 | 0.01 | 76.0 | 50.8 |
| ViT-B/16 | 0.33 | 96.4 | 52.9 |
| ViT-B/16-in21k | 0.20 | 92.1 | 51.3 |
| ViT-L/16 | 0.26 | 96.1 | 52.9 |
| ViT-S/16-DINO | 0.48 | 97.9 | 55.5 |
| ViT-B/16-DINO | 0.55 | 99.3 | 50.4 |
| ViT-S/8-DINO | 0.40 | 95.8 | 51.1 |
| ViT-B/8-DINO | 0.50 | 98.8 | 49.6 |
| Res50-SimCLRv2 | 0.66 | 99.8 | 50.1 |
| Res101-SimCLRv2 | 0.60 | 99.4 | 51.6 |

when we vary $\epsilon$ (cf. Appendix Table 6). Similar conclusions can also be drawn by referring to self-supervised pretrained representations, rows "ViT-S/-DINO" and "ViT-B/-DINO". Moreover, if we check rows "ViT-B/16" and "ViT-B/16-DINO", we compare two pretrained models of the same architecture but trained under different regimes, either supervised or self-supervised. Between these two models, SynBench favors self-supervised trained "ViT-B/16-DINO", echoing with the inductive bias of self-supervised contrastive learning discovered in recent literature (HaoChen & Ma, 2022).

**SynBench shows better correlation with real-data probing accuracy and robustness.** We run baselines as described in Section 4.1 for the synthetic classification task on pretrained models with dataset size $n$ being $2048, 8192, 32768$ and list their results in Appendix Table 7. Throughout our experiments, we use 2048 test samples in the synthetic dataset. For Val loss, MDL, and SDL, $\epsilon$SC, the smaller the better; for LogME, SFDA, SynBench, the bigger the better. We calculate the correlation between task-agnostic evaluation metrics and real-life data tasks as a function of the dataset size $n$ in

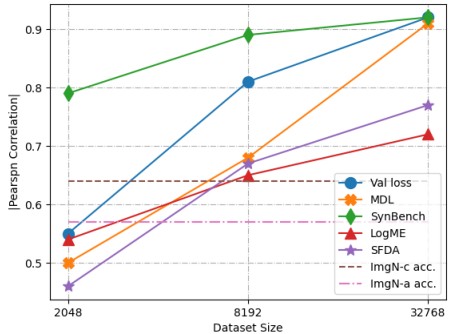

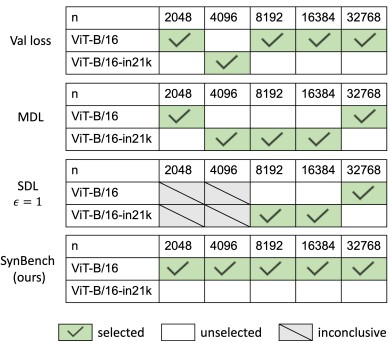

Figure 4: Pearson correlation between task-agnostic metrics (Val loss, MDL, SynBench, LogME, SFDA) and task-specific metrics (the average accuracy on 27 real-life tasks) as functions of the dataset size. Two dashed lines characterize the correlation by transfer datasets' accuracy.

Figure 5: Comparison of model selections using task-agnostic benchmarks. We denote the model predicted to have better performance by "selected". Only SynBench gives consistent selections across varying data sample sizes. Refer to Appendix Table 8 for more details.

Table 2: CIFAR10 and TinyImagenet standard and robust accuracy (%) changes ($\delta$SA and $\delta$RA) using $\epsilon$-robust linear probing ($\epsilon$-robust prob.). We see that $\epsilon$-robust prob. with $\epsilon = \arg\max_\epsilon$SynBench-Score gives the best robust accuracy.

| Models | | CIFAR10 | | | | TinyImagenet | | | |
|---|---|---|---|---|---|---|---|---|---|
| | | $\epsilon = 0$ | $\epsilon = 0.1$ | $\epsilon = 0.2$ | $\epsilon = 0.3$ | $\epsilon = 0$ | $\epsilon = 0.1$ | $\epsilon = 0.2$ | $\epsilon = 0.3$ |
| ViT-Ti/16 | SynBench-Score($\epsilon$) | **0.01** | **0.01** | 0 | 0 | **0.01** | **0.01** | 0 | 0 |
| | $\epsilon$-robust prob. $\delta$SA | 0 | -3.1 | -5.9 | -6.3 | 0 | +0.3 | -1.5 | -1.9 |
| | $\epsilon$-robust prob. $\delta$RA | 0 | +1.4 | +1.9 | +1.6 | 0 | +1.1 | +0.4 | +2.2 |
| ViT-B/16 | SynBench-Score($\epsilon$) | 0.33 | 0.36 | **0.37** | 0.35 | 0.33 | 0.36 | **0.37** | 0.35 |
| | $\epsilon$-robust prob. $\delta$SA | 0 | +0.2 | +0.1 | +0.1 | 0 | 0 | +0.7 | +0.6 |
| | $\epsilon$-robust prob. $\delta$RA | 0 | +0.3 | +2.7 | +2.3 | 0 | -1.0 | +2.5 | +2.4 |
| ViT-B/16-in21k | SynBench-Score($\epsilon$) | 0.20 | 0.22 | **0.23** | 0.21 | 0.20 | 0.22 | **0.23** | 0.21 |
| | $\epsilon$-robust prob. $\delta$SA | 0 | +0.9 | +1.1 | +1.1 | 0 | +0.3 | +0.3 | +0.2 |
| | $\epsilon$-robust prob. $\delta$RA | 0 | +1.2 | +1.4 | +0.6 | 0 | +1.3 | +2.0 | +2.0 |
| ViT-L/16 | SynBench-Score($\epsilon$) | 0.26 | 0.30 | **0.33** | 0.32 | 0.26 | 0.30 | **0.33** | 0.32 |
| | $\epsilon$-robust prob. $\delta$SA | 0 | +0.2 | +0.4 | +0.4 | 0 | -0.1 | -0.2 | -0.3 |
| | $\epsilon$-robust prob. $\delta$RA | 0 | -0.2 | +3.0 | +1.9 | 0 | +4.2 | +6.6 | +0.7 |

Figure 4. Specifically, we calculate the Pearson correlation coefficients between the average accuracy in downstream tasks to scores given by Val loss, MDL, SDL, $\epsilon$SC, LogME, SFDA, and SynBench (SDL and $\epsilon$SC are excluded from the figure since they fail to give concrete numbers for small dataset sizes). With 2k synthetic samples, SynBench gives 0.79, whereas Val loss, MDL, LogME, and SFDA range between 0.46 and 0.55; with 8k synthetic samples, SynBench gives 0.89, whereas Val loss, MDL, LogME, and SFDA range between 0.65 and 0.81, surpassing the correlation by vanilla out-of-distribution accuracy (ImageNet-c's 0.64 and ImageNet-a's 0.57); with over 30k synthetic samples, Val loss, MDL, and SynBench all indicate very strong correlation ($> 0.9$) with real-life data accuracy, confirming the feasibility of probing pretrained representations in a task-agnostic yet effective way. To validate the capability of SynBench in informing model robustness, we further conduct CW attack Carlini & Wagner (2017), on CIFAR10 test set and calculate its correlation with SynBench. With 2k, 8k, and 30k synthetic samples, SynBench is also able to demonstrate moderate correlation with coefficient ranging from 0.74 to 0.84.

**SynBench gives more consistent suggestions than baselines.** We run a finer grid on the dataset size $n \in \{2048, 4096, 8192, 16384, 32768\}$ and compare the consistency of each metrics. Since LogME and SFDA showed worse correlation in the previous experiment, we exclude the two and only report the results on Val loss, MDL, and SynBench. We also include SDL to highlight its struggle with small sample size. In Figure 5, we give an example of the model selections between ViT-B/16 and ViT-B/16-in21k. Detailed numbers are reported in Appendix Table 8. It is worth noting that SynBench consistently recommends ViT-B/16 over ViT-B/16-in21k, while other methods change with $n$. Besides better correlation and consistency, the runtime analysis in Appendix A.8 also confirms $50\times$ speedup over baselines using SynBench.

### 4.3 SynBench-informed Robust Linear Probing

When fine-tuning a linear probing layer on popular CV downstream datasets (e.g. CIFAR10, TinyImageNet), one can implement $\epsilon$-robust linear probing for better robustness (Fan et al., 2021). Concretely, let $\theta$ be the pretrained representation network and $\theta_c$ be the probing layer parameters, $\epsilon$-robust linear

Table 3: Task-specific linear probing standard accuracy and robust accuracy (%).

| Models | CIFAR10 SA | CIFAR10 RA | SVHN SA | SVHN RA | TinyImageNet SA | TinyImageNet RA |
|--------|------|------|------|------|------|------|
| ViT-Ti/16 | 81.9 | 1.1 | 48.0 | 0.7 | 42.93 | 3.36 |
| ViT-B/16 | 95.0 | 32.1 | 65.4 | 5.2 | 74.65 | 33.67 |
| ViT-L/16 | 98.0 | 57.0 | 68.9 | 8.4 | 86.58 | 55.0 |

Table 4: SynBench-Scores on synthetic data with heptadiagonal covariance (Gaussian-H).

| Models | $\epsilon = 0$ | $\epsilon = 0.2$ | $\epsilon = 0.4$ | $\epsilon = 0.6$ | $\epsilon = 0.8$ |
|--------|------|------|------|------|------|
| ViT-Ti/16 | 0 | 0 | 0 | 0 | 0 |
| ViT-B/16 | 0.18 | 0.24 | 0.20 | 0.10 | 0.01 |
| ViT-L/16 | 0.18 | 0.28 | 0.28 | 0.23 | 0.12 |

probing solves $\min_{\theta_c} \max_{\delta: \|\delta\|_2 \le \epsilon} \mathbb{E}_{(x,y) \in \mathcal{D}} \ell_{\text{Cross-entropy}}(f_{\theta_c} \circ f_\theta(x+\delta), y)$. Here we will demonstrate how SynBench-Scores help to select $\epsilon$ without accessing the downstream data.

In Table 1, we only give SynBench-Scores with $\epsilon = 0$. We refer readers to Appendix Table 6 for the full table with different $\epsilon$. We cite 4 pretrained representations' SynBench-Score in Table 2 and observe that, for each model, SynBench-score is not necessarily monotonic in $\epsilon$ (peaks are boldfaced). For example, the SynBench-Score for ViT-B/16 peaks at $\epsilon = 0.2$, which indicates standard linear probing (i.e., $\epsilon = 0$) may not be the most effective way to probe pretrained representations in terms of robustness-accuracy performance. This interesting indication is consistent with recent findings (Fan et al., 2021). We hereby implement $\epsilon$-robust linear probing and verify $\epsilon = \arg\max_\epsilon \text{SynBench-Score}$ can indeed yield a better robustness-accuracy tradeoff in Table 2. The robust accuracy herein is obtained by AutoAttack (Croce & Hein, 2020) with attack strength 0.2. From the table, we see that SynBench-informed $\epsilon$-robust linear probing does find the best overall robustness and accuracy. For instance, SynBench-Score peaks at $\epsilon = 0.2$ for ViT-B/16 and correspondingly 0.2-robust linear probing on ViT-B/16 representations improve the standard and robust accuracy the most (+0.1%/+0.7% and +2.7%/+2.5%).

## 4.4 THE EFFECT OF DATA PRIOR

In Section 3.4, it is stated that a more precise capture of the pretrained representation performance can be given if one has some prior knowledge of the downstream data distribution. In this section, we show this point by studying three specific downstream tasks, CIFAR10, SVHN, and TinyImageNet classifications, and give an example of the devised covariance matrix for SynBench synthetic Gaussians. In Table 3, we give the standard and robust accuracy on CIFAR10, SVHN, and TinyImageNet (robust accuracy obtained by AutoAttack). Comparing the rows "ViT-B/16" and "ViT-L/16", it is observed that ViT-L/16 is in fact performing better than ViT-B/16 on these three downstream tasks, whereas SynBench-Score with identity covariance suggests the opposite (cf. Table 1). To uncover the reason behind the inconsistency, we calculate the distance between the synthetic Gaussian used throughout the experiments till now (dubbed Gaussian-I) and these datasets in Appendix Table 11. Recall that Gaussian-I, $P_{\mu_1,\mu_2,\Sigma}$, has $\mu_1 = -\mu_2 = s_i \cdot 1_d/\sqrt{d}$ and $\Sigma = I_d$. An easy modification on the covariance matrix $\Sigma$ leads us to Gaussian-H, $P_{\mu_1,\mu_2,\Sigma}$ with $\mu_1 = -\mu_2 = s_i \cdot 1_d/\sqrt{d}$ and $\Sigma$ be a channel-wise band matrix covariance. Gaussian-H captures the case when the R,G,B channel entries are externally independent (hence overall a block-diagonal covariance matrix with each of the 3 blocks being $224^2 \times 224^2$), and internally correlated based on locality (each block is a heptadiagonal matrix where only the main diagonal, and the first three diagonals above and below it have nonzero entries). Note that Gaussian-H is closer to the three datasets compared to Gaussian-I with respect to Fréchet inception distance (FID) (Heusel et al., 2017) and Mahalanobis distance (MD) (Mahalanobis, 1936) according to Table 11. Based on Gaussian-H, SynBench now recommends ViT-L/16 over ViT-B/16 according to Table 4. We defer more results with Gaussian-H covariate synthetic data to Appendix Table 12-14. This result shows that SynBench can incorporate complex data structures and downstream data characteristics into the process of synthetic data generation.

## 5 DISCUSSION AND CONCLUSION

In this paper, we proposed a new **task-agnostic** framework *SynBench* for benchmarking the robustness-accuracy performance of pretrained representations. SynBench is fundamentally task-independent and provides a quantifiable score that does not rely on any real-life data. SynBench exploits an idealized data distribution, class conditional Gaussian mixture, to establish a theoretically-derived robustness-accuracy tradeoff, which serves as the reference for pretrained representations. Finally, a quantifiable score *SynBench-Score* was provided that compares the ratio of area-under-the-curve between the reference and the pretrained representations. We validated the usefulness of SynBench-Score on several pretrained vision models in giving insightful comparisons on model attributes. We demonstrated its high correlation with real-life tasks, and showed its consistent model selections. While delving into the robustness-accuracy characterization of pretrained representations, we envision the SynBench framework to be further extended to other trustworthiness dimensions (e.g., privacy and fairness) and other domains, to shed light on task-agnostic benchmarking designs.

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

# A APPENDIX

## A.1 USAGE

SynBench offers a pricipled evaluation framework that serves as a sanity representation quality check for pretrained models. We choose the idealized Gaussian mixtures for evaluating pre-trained models because it is shown to be capable of modeling the statistics of natural images (Zoran & Weiss, 2012) and its optimial robustness-accuracy tradeoff can be analytically chatacterized based on our Theorem 1. We view SynBench as a "necessary" and "minimum" model test in the sense that, with perfect data sampled from an ideal distribution, any undesirable deteriorated behavior (such as weakened robustness) reveals the weaknesses of the representation model that could possibly lead to vulnerabilities in real-life downstream tasks. Therefore, in designing this minimum test, it is important that the task has a theoretical ideal (and optimal) solution (i.e. the trade-off preserved by class conditional Gaussians, Theorem 1 iv).

Here are some possible scenarios to use our developed tool:

- model auditing: use SynBench to generate diverse psuedo tasks (e.g., with diffrent difficulty levels) and compare them with theoretically optimial results, for a comprehensive evaluation on the capability of a pre-trained model

- hyperparameter tuning: as shown in Sec. 4.3, SynBench can be used for hyperparameter selection in robust linear probing, which leads to improved performance in the considered downstream tasks.

- model selection (without using downstream data): without the knowledge of downstream applications, one can use SynBench to rank the quality of pre-trained representations (e.g., the example shown in Figure 4). It is also possible to incorporate some known statistics of the downstream dataset into guided synthetic data generaltion and evaluation in SynBench, as discussed in Sec. 4.4.

- model training: while updating a model in the pre-training state, one can use SynBench to ensure the model performance (in terms of SynBench-Score) is aligned.

## A.2 OBJECTIVE

$$
\begin{aligned}
E_{\theta,\epsilon}(a_t) =& \mathbb{E}_{s \sim \mathcal{U}, x - \bar{\mu} | y \sim \mathcal{N}(\mu, \Sigma)} \left[ \|\bar{\Delta}\|_2 \mid f_\epsilon(x) = y, a > a_t, \mu = s \cdot 1_d / \sqrt{d}, \Sigma = I_d \right] \\
=& \mathbb{E}_{s,x} \left[ \|\bar{\Delta}\|_2 \mid f_\epsilon(x) = y, a(s, \epsilon) > a_t \right] \\
=& \sum_i \mathbb{E}_x \left[ \|\bar{\Delta}\|_2 \mid f_\epsilon(x) = y, a(s_i, \epsilon) > a_t \right] \mathbb{P}(s = s_i) \\
=& \frac{1}{n} \sum_i \mathbb{E}_x \left[ \|\bar{\Delta}\|_2 \mid f_\epsilon(x) = y, a(s_i, \epsilon) > a_t \right] \\
=& \frac{1}{n} \sum_i \mathbb{E}_x \left[ \|\bar{\Delta}\|_2 \mid f_\epsilon(x) = y \right] \mathbb{1}_{a(s_i, \epsilon) > a_t}.
\end{aligned}
$$

## A.3 PROOFS

**Theorem 2.** *For any sample $x$, the optimal robust classifier $f_\epsilon$ for $P_{\mu_1, \mu_2, \Sigma}$ gives*

(i) *the bound (decision margin)* $\|\Delta\|_2 = \frac{|(x - \frac{\mu_1 + \mu_2}{2})^T \Sigma^{-1}(\tilde{\mu} - z_\Sigma(\tilde{\mu}))|}{\|\Sigma^{-1}(\tilde{\mu} - z_\Sigma(\tilde{\mu}))\|_2}$,

(ii) *the scaled bound* $\|\bar{\Delta}\|_2 = \frac{|(x - \frac{\mu_1 + \mu_2}{2})^T \Sigma^{-1}(\tilde{\mu} - z_\Sigma(\tilde{\mu}))|}{|\tilde{\mu}^T \Sigma^{-1}(\tilde{\mu} - z_\Sigma(\tilde{\mu}))|}$.

*For a sample $x \sim P_{\mu_1, \mu_2, \Sigma}$, it further gives*

(iii) *the standard accuracy* $a = \Phi\left(\frac{\tilde{\mu}^T \Sigma^{-1}(\tilde{\mu} - z_\Sigma(\tilde{\mu}))}{\|\Sigma^{-1}(\tilde{\mu} - z_\Sigma(\tilde{\mu}))\|_\Sigma}\right)$,

*(iv) the expected scaled bound of correct samples* $\mathbb{E}\left[\|\bar{\Delta}\|_2 \mid f_\epsilon(x) = y\right] = \frac{1}{\sqrt{2\pi}}\frac{1}{a\Phi^{-1}(a)}e^{-\frac{1}{2}\left(\Phi^{-1}(a)\right)^2} + 1,$

*where $z_\Sigma$ is the solution of the convex problem $\arg\min_{\|z\|_2 \le \epsilon}(\tilde{\mu} - z)^T\Sigma^{-1}(\tilde{\mu} - z)$ and $\Phi$ denotes the CDF of the standard normal distribution.*

*Proof.* (i) Following Bhagoji et al. (2019); Dan et al. (2020), the Bayes optimal robust classifier for the general non-symmetric conditional Gaussians $P_{\mu_1,\mu_2,\Sigma}$ specified in equation 1 is

$$f_\epsilon(x) = sign\left\{\left(x - \frac{\mu_1 + \mu_2}{2}\right)^T \Sigma^{-1}\left(\tilde{\mu} - z_\Sigma(\tilde{\mu})\right)\right\}, \tag{4}$$

where $sign(\cdot)$ is the typical sign function and $z_\Sigma$ is the solution of the convex problem $\arg\min_{\|z\|_2 \le \epsilon}(\tilde{\mu} - z)^T\Sigma^{-1}(\tilde{\mu} - z)$. The corresponding decision boundary is at $\left((x + \delta) - \frac{\mu_1+\mu_2}{2}\right)^T \Sigma^{-1}\left(\tilde{\mu} - z_\Sigma(\tilde{\mu})\right) = 0,$

$$\implies \quad \Delta = \arg\min \|\delta\|_2 \quad \text{s.t.} \quad \delta^T\Sigma^{-1}\left(\tilde{\mu} - z_\Sigma(\tilde{\mu})\right) = -\left(x - \frac{\mu_1 + \mu_2}{2}\right)^T\Sigma^{-1}\left(\tilde{\mu} - z_\Sigma(\tilde{\mu})\right)$$

$$\implies \quad \|\Delta\|_2 = \frac{|(x - \frac{\mu_1+\mu_2}{2})^T\Sigma^{-1}\left(\tilde{\mu} - z_\Sigma(\tilde{\mu})\right)|}{\|\Sigma^{-1}\left(\tilde{\mu} - z_\Sigma(\tilde{\mu})\right)\|_2}.$$

(ii) Since the bound $\|\Delta\|_2$ is subject to the positions of two Gaussians, we scale the bound by the distance from Gaussian centers to the classifier, $\frac{|\tilde{\mu}^T\Sigma^{-1}(\tilde{\mu} - z_\Sigma(\tilde{\mu}))|}{\|\Sigma^{-1}(\tilde{\mu} - z_\Sigma(\tilde{\mu}))\|_2}$ and obtain

$$\|\bar{\Delta}\|_2 = \frac{|(x - \frac{\mu_1+\mu_2}{2})^T\Sigma^{-1}(\tilde{\mu} - z_\Sigma(\tilde{\mu}))|}{\|\Sigma^{-1}(\tilde{\mu} - z_\Sigma(\tilde{\mu}))\|_2}\frac{\|\Sigma^{-1}(\tilde{\mu} - z_\Sigma(\tilde{\mu}))\|_2}{|\tilde{\mu}^T\Sigma^{-1}(\tilde{\mu} - z_\Sigma(\tilde{\mu}))|}$$

$$= \frac{|(x - \frac{\mu_1+\mu_2}{2})^T\Sigma^{-1}(\tilde{\mu} - z_\Sigma(\tilde{\mu}))|}{|\tilde{\mu}^T\Sigma^{-1}(\tilde{\mu} - z_\Sigma(\tilde{\mu}))|}.$$

(iii) For sample $x \sim P_{\mu_1,\mu_2,\Sigma}$, consider the Bayes optimal robust classifier in equation 4, we can calculate the analytical standard accuracy by

$$\mathbb{P}(y = 1)\mathbb{P}\left[f_\epsilon(x) = 1 \mid y = 1\right] + \mathbb{P}(y = -1)\mathbb{P}\left[f_\epsilon(x) = -1 \mid y = -1\right]$$
$$= \mathbb{P}\left[f_\epsilon(x) = 1 \mid y = 1\right]$$
$$= \mathbb{P}\left[(x - \frac{\mu_1 + \mu_2}{2})^T\Sigma^{-1}(\tilde{\mu} - z_\Sigma(\tilde{\mu})) > 0 \mid y = 1\right]$$
$$= \mathbb{P}\left[(\tilde{\mu} + w)^T\Sigma^{-1}(\tilde{\mu} - z_\Sigma(\tilde{\mu})) > 0\right], \quad w \sim \mathcal{N}(0, \Sigma)$$
$$= \mathbb{P}\left[w^T\Sigma^{-1}(\tilde{\mu} - z_\Sigma(\tilde{\mu})) > -\tilde{\mu}^T\Sigma^{-1}(\tilde{\mu} - z_\Sigma(\tilde{\mu}))\right], \quad w \sim \mathcal{N}(0, \Sigma)$$
$$= \mathbb{P}\left[\frac{w^T\Sigma^{-1}(\tilde{\mu} - z_\Sigma(\tilde{\mu}))}{\|\Sigma^{-1}(\tilde{\mu} - z_\Sigma(\tilde{\mu}))\|_\Sigma} > -\frac{\tilde{\mu}^T\Sigma^{-1}(\tilde{\mu} - z_\Sigma(\tilde{\mu}))}{\|\Sigma^{-1}(\tilde{\mu} - z_\Sigma(\tilde{\mu}))\|_\Sigma}\right], \quad \frac{w^T\Sigma^{-1}(\tilde{\mu} - z_\Sigma(\tilde{\mu}))}{\|\Sigma^{-1}(\tilde{\mu} - z_\Sigma(\tilde{\mu}))\|_\Sigma} \sim \mathcal{N}(0, 1)$$
$$= \Phi\left(\frac{\tilde{\mu}^T\Sigma^{-1}(\tilde{\mu} - z_\Sigma(\tilde{\mu}))}{\|\Sigma^{-1}(\tilde{\mu} - z_\Sigma(\tilde{\mu}))\|_\Sigma}\right).$$

(iv) For sample $x \sim P_{\mu_1,\mu_2,\Sigma}$, let $a$ denote the accuracy, $t$ denote $x - \frac{\mu_1+\mu_2}{2}$, and $w$ denote $\Sigma^{-1}(\tilde{\mu} - z_\Sigma(\tilde{\mu}))$. From (iii), we have that the standard accuracy of conditional Gaussian samples with the Bayes optimal (robust) classifier is $\Phi(\frac{\tilde{\mu}^T w}{\|w\|_\Sigma})$, so $\frac{\tilde{\mu}^T w}{\|w\|_\Sigma} = \Phi^{-1}(a)$. Since for binary classification, we only care about accuracy from 0.5 to 1, so we should have $\tilde{\mu}^T w > 0$.

Now consider the classifier in equation 4 and the corresponding scaled bound from (ii),

$$\|\bar{\Delta}\|_2 = \frac{|(x - \frac{\mu_1+\mu_2}{2})^T\Sigma^{-1}(\tilde{\mu} - z_\Sigma(\tilde{\mu}))|}{|\tilde{\mu}^T\Sigma^{-1}(\tilde{\mu} - z_\Sigma(\tilde{\mu}))|} = \frac{|t^T w|}{|\tilde{\mu}^T w|} = \frac{|t^T w|}{\tilde{\mu}^T w}.$$

Since $t|y \sim \mathcal{N}(y\tilde{\mu}, \Sigma)$, we have $t^T w|y \sim \mathcal{N}(y\tilde{\mu}^T w, w^T \Sigma^T w)$. When we only want to get the expected scaled bound of the correctly-classified samples, we have that

$$
\begin{aligned}
\mathbb{E}\left[\|\bar{\Delta}\|_2 \mid f_\epsilon(x) = y\right] &= \frac{1}{\tilde{\mu}^T w}\mathbb{E}\left[|t^T w| \mid f_\epsilon(x) = y\right] \\
&= \frac{1}{2\tilde{\mu}^T w}\mathbb{E}\left[|t^T w| \mid f_\epsilon(x) = y = 1\right] + \frac{1}{2\tilde{\mu}^T w}\mathbb{E}\left[|t^T w| \mid f_\epsilon(x) = y = -1\right] \\
&= \frac{1}{2\tilde{\mu}^T w}\mathbb{E}\left[t^T w \mid y = 1, t^T w \geq 0\right] + \frac{1}{2\tilde{\mu}^T w}\mathbb{E}\left[-t^T w \mid y = -1, t^T w < 0\right].
\end{aligned}
$$

Recall that $t^T w|y \sim \mathcal{N}(y\tilde{\mu}^T w, w^T \Sigma^T w)$, then by the mean of truncated normal distribution, it is true that

$$
\begin{aligned}
\mathbb{E}\left[t^T w \mid y = 1, t^T w \geq 0\right] &= \tilde{\mu}^T w + \sqrt{w^T \Sigma^T w}\frac{\phi(\frac{0 - \tilde{\mu}^T w}{\sqrt{w^T \Sigma^T w}})}{1 - \Phi(\frac{0 - \tilde{\mu}^T w}{\sqrt{w^T \Sigma^T w}})} \\
&= \tilde{\mu}^T w + \sqrt{w^T \Sigma^T w}\frac{\phi(-\frac{\tilde{\mu}^T w}{\sqrt{w^T \Sigma^T w}})}{1 - \Phi(-\frac{\tilde{\mu}^T w}{\sqrt{w^T \Sigma^T w}})} \\
&= \tilde{\mu}^T w + \sqrt{w^T \Sigma^T w}\frac{1}{\sqrt{2\pi}\Phi(\frac{\tilde{\mu}^T w}{\sqrt{w^T \Sigma^T w}})}e^{-\frac{1}{2}\left(\frac{\tilde{\mu}^T w}{\sqrt{w^T \Sigma^T w}}\right)^2} \\
\mathbb{E}\left[-t^T w \mid y = -1, t^T w < 0\right] &= -\mathbb{E}\left[t^T w \mid y = -1, t^T w < 0\right] \\
&= -\left(-\tilde{\mu}^T w - \sqrt{w^T \Sigma^T w}\frac{\phi(\frac{0 + \tilde{\mu}^T w}{\sqrt{w^T \Sigma^T w}})}{\Phi(\frac{0 + \tilde{\mu}^T w}{\sqrt{w^T \Sigma^T w}})}\right) \\
&= \tilde{\mu}^T w + \sqrt{w^T \Sigma^T w}\frac{1}{\sqrt{2\pi}\Phi(\frac{\tilde{\mu}^T w}{\sqrt{w^T \Sigma^T w}})}e^{-\frac{1}{2}\left(\frac{\tilde{\mu}^T w}{\sqrt{w^T \Sigma^T w}}\right)^2}.
\end{aligned}
$$

Therefore

$$
\begin{aligned}
\mathbb{E}\left[\|\bar{\Delta}\|_2 \mid f_\epsilon(x) = y\right] &= \frac{1}{\tilde{\mu}^T w}\left(\tilde{\mu}^T w + \sqrt{w^T \Sigma^T w}\frac{1}{\sqrt{2\pi}\Phi(\frac{\tilde{\mu}^T w}{\sqrt{w^T \Sigma^T w}})}e^{-\frac{1}{2}\left(\frac{\tilde{\mu}^T w}{\sqrt{w^T \Sigma^T w}}\right)^2}\right) \\
&= 1 + \frac{\sqrt{w^T \Sigma^T w}}{\tilde{\mu}^T w}\frac{1}{\sqrt{2\pi}\Phi(\frac{\tilde{\mu}^T w}{\sqrt{w^T \Sigma^T w}})}e^{-\frac{1}{2}\left(\frac{\tilde{\mu}^T w}{\sqrt{w^T \Sigma^T w}}\right)^2}.
\end{aligned}
$$

By replacing $\frac{\tilde{\mu}^T w}{\sqrt{w^T \Sigma^T w}}$ by $\Phi^{-1}(a)$, we got

$$
\mathbb{E}\left[\|\bar{\Delta}\|_2 \mid f_\epsilon(x) = y\right] = \frac{1}{\sqrt{2\pi}}\frac{1}{a\Phi^{-1}(a)}e^{-\frac{1}{2}\left(\Phi^{-1}(a)\right)^2} + 1.
$$

$\square$

### A.4 GENERAL $\ell_p$ RESULTS

We note that our results in Appendix A.3 can be straightforwardly generalized to $\ell_p$. Given an $\ell_p$ adversarial budget $\epsilon$:

**Theorem 3.** *For any sample $x$, the optimal robust classifier $f_\epsilon$ for $P_{\mu_1,\mu_2,\Sigma}$ gives*

(i) *the bound (decision margin)* $\|\Delta\|_p = \frac{|(x - \frac{\mu_1 + \mu_2}{2})^T \Sigma^{-1}(\tilde{\mu} - z_\Sigma(\tilde{\mu}))|}{\|\Sigma^{-1}(\tilde{\mu} - z_\Sigma(\tilde{\mu}))\|_q}$,

(ii) *the scaled bound* $\|\bar{\Delta}\|_p = \frac{|(x - \frac{\mu_1 + \mu_2}{2})^T \Sigma^{-1}(\tilde{\mu} - z_\Sigma(\tilde{\mu}))|}{|\tilde{\mu}^T \Sigma^{-1}(\tilde{\mu} - z_\Sigma(\tilde{\mu}))|}$.

*For sample $x \sim P_{\mu_1,\mu_2,\Sigma}$, it further gives*

(iii) the standard accuracy $a = \Phi\left(\frac{\tilde{\mu}^T \Sigma^{-1}(\tilde{\mu} - z_\Sigma(\tilde{\mu}))}{\|\Sigma^{-1}(\tilde{\mu} - z_\Sigma(\tilde{\mu}))\|_\Sigma}\right)$,

(iv) the expected scaled bound of correct samples $\mathbb{E}\left[\|\bar{\Delta}\|_p \mid f_\epsilon(x) = y\right] = \frac{1}{\sqrt{2\pi}} \frac{1}{a \Phi^{-1}(a)} e^{-\frac{1}{2}\left(\Phi^{-1}(a)\right)^2} + 1$,

where $z_\Sigma$ is the solution of the convex problem $\arg\min_{\|z\|_p \leq \epsilon}(\tilde{\mu} - z)^T \Sigma^{-1}(\tilde{\mu} - z)$ and $\Phi$ denotes the CDF of the standard normal distribution.

*Proof.* We follow the proof of Theorem 1 and consider the classifier in equation 4. By Hölder's inequality, we now have the corresponding lower bound and scaled lower bound as

$$\|\Delta\|_p = \frac{|(x - \frac{\mu_1 + \mu_2}{2})^T \Sigma^{-1}(\tilde{\mu} - z_\Sigma(\tilde{\mu}))|}{\|\Sigma^{-1}(\tilde{\mu} - z_\Sigma(\tilde{\mu}))\|_q}$$

$$\|\bar{\Delta}\|_p = \frac{|(x - \frac{\mu_1 + \mu_2}{2})^T \Sigma^{-1}(\tilde{\mu} - z_\Sigma(\tilde{\mu}))|}{\|\Sigma^{-1}(\tilde{\mu} - z_\Sigma(\tilde{\mu}))\|_q} \frac{\|\Sigma^{-1}(\tilde{\mu} - z_\Sigma(\tilde{\mu}))\|_q}{|\tilde{\mu}^T \Sigma^{-1}(\tilde{\mu} - z_\Sigma(\tilde{\mu}))|}$$

$$= \frac{|(x - \frac{\mu_1 + \mu_2}{2})^T \Sigma^{-1}(\tilde{\mu} - z_\Sigma(\tilde{\mu}))|}{|\tilde{\mu}^T \Sigma^{-1}(\tilde{\mu} - z_\Sigma(\tilde{\mu}))|},$$

where $\frac{1}{p} + \frac{1}{q} = 1$. The remainder of the proof will then follows as in Theorem 1. $\square$

**Remark.** In general, in the case that $\Sigma$ is singular, we can apply the economy-size (thin) decomposition with nonzero eigenvalues $\Sigma = F\Lambda F^T$. Then, with a general non-symmetric conditional Gaussians

$$x|y = 1 \sim \mathcal{N}(\mu_1, \Sigma), \quad x|y = -1 \sim \mathcal{N}(\mu_2, \Sigma),$$

we apply proper translation to symmetric conditional Gaussians

$$F^T x|y = 1 \sim \mathcal{N}(F^T \mu_1, \Lambda), \quad F^T x|y = -1 \sim \mathcal{N}(F^T \mu_2, \Lambda),$$

$$F^T x - F^T \frac{\mu_1 + \mu_2}{2}|y = 1 \sim \mathcal{N}(\tilde{\mu}, \Lambda), \quad F^T x - F^T \frac{\mu_1 + \mu_2}{2}|y = -1 \sim \mathcal{N}(-\tilde{\mu}, \Lambda),$$

where $\tilde{\mu} = F^T \frac{\mu_1 - \mu_2}{2}$.

## A.5 CLASS IMBALANCE

Given an $\ell_2$ adversarial budget $\epsilon \leq \|\mu\|_2$, consider the conditional Gaussian in equation 1 with $\Sigma = I_d$ ($d$ by $d$ identity matrix) and general class prior $\tau$, then the following theorem holds.

**Theorem 4.** *For any sample $x$, the optimal robust classifier $f_\epsilon$ for $P_{\mu_1, \mu_2, I_d}$ gives*

(i) *the bound (decision margin)* $\|\Delta\|_2 = \frac{|(x - \frac{\mu_1 + \mu_2}{2})^T \tilde{\mu}(1 - \epsilon/\|\tilde{\mu}\|_2) - q/2|}{\|\tilde{\mu}(1 - \epsilon/\|\tilde{\mu}\|_2)\|_2}$,

(ii) *the scaled bound* $\|\bar{\Delta}\|_2 = \frac{2|(x - \frac{\mu_1 + \mu_2}{2})^T \tilde{\mu}(1 - \epsilon/\|\tilde{\mu}\|_2) - q/2|}{|\tilde{\mu}^T \tilde{\mu}(1 - \epsilon/\|\tilde{\mu}\|_2) - q/2| + |\tilde{\mu}^T \tilde{\mu}(1 - \epsilon/\|\tilde{\mu}\|_2) + q/2|}$.

*For a sample $x \sim P_{\mu_1, \mu_2, I_d}$, it further gives*

(iii) *the standard accuracy* $a = \tau\Phi\left(\frac{\tilde{\mu}^T w - q/2}{\|w\|_2}\right) + (1 - \tau)\Phi\left(\frac{\tilde{\mu}^T w + q/2}{\|w\|_2}\right)$,

(iv) *the expected scaled bound of correct samples*

$$\mathbb{E}\left[\|\bar{\Delta}\|_2 \mid f_\epsilon(x) = y\right] = \frac{2\tau}{|\tilde{\mu}^T w - q/2| + |\tilde{\mu}^T w + q/2|}\left(\tilde{\mu}^T w - q/2 + \|w\|_2 \frac{\phi\left(\frac{-\tilde{\mu}^T w + q/2}{\|w\|_2}\right)}{\Phi\left(\frac{\tilde{\mu}^T w - q/2}{\|w\|_2}\right)}\right)$$

$$+ \frac{2(1 - \tau)}{|\tilde{\mu}^T w - q/2| + |\tilde{\mu}^T w + q/2|}\left(\tilde{\mu}^T w + q/2 + \|w\|_2 \frac{\phi\left(\frac{\tilde{\mu}^T w + q/2}{\|w\|_2}\right)}{\Phi\left(\frac{\tilde{\mu}^T w + q/2}{\|w\|_2}\right)}\right).$$

where $q = ln\{(1-\tau)/\tau\}$, $w = \tilde{\mu}(1 - \epsilon/\|\tilde{\mu}\|_2)$, $\phi$ and $\Phi$ denotes the PDF and CDF of the standard normal distribution.

*Proof.* (i) Consider the Bayes optimal $\ell_2$ $\epsilon$-robust classifier (Dobriban et al., 2020, Theorem 4.1)

$$f_\epsilon(x) = sign\left\{ \left(x - \frac{\mu_1 + \mu_2}{2}\right)^T \tilde{\mu}(1 - \epsilon/\|\tilde{\mu}\|_2) - q/2 \right\}, \tag{5}$$

where $q = ln\{(1-\tau)/\tau\}$. For any $x$,

$$\|\Delta\|_2 = \frac{|(x - \frac{\mu_1+\mu_2}{2})^T \tilde{\mu}(1 - \epsilon/\|\tilde{\mu}\|_2) - q/2|}{\|\tilde{\mu}(1 - \epsilon/\|\tilde{\mu}\|_2)\|_2}.$$

(ii) Since the bound $\|\Delta\|_2$ is subject to the positions of two Gaussians, we scale the bound by the distance from Gaussian centers to the classifier. We note that now the distances from the two Gaussian centers to the classifier are different, $\frac{|\tilde{\mu}^T\tilde{\mu}(1-\epsilon/\|\tilde{\mu}\|_2)-q/2|}{\|\tilde{\mu}(1-\epsilon/\|\tilde{\mu}\|_2)\|_2}$ and $\frac{|\tilde{\mu}^T\tilde{\mu}(1-\epsilon/\|\tilde{\mu}\|_2)+q/2|}{\|\tilde{\mu}(1-\epsilon/\|\tilde{\mu}\|_2)\|_2}$. We hereby take their average as the scaling factor and obtain

$$\|\bar{\Delta}\|_2 = \frac{|(x - \frac{\mu_1+\mu_2}{2})^T \tilde{\mu}(1 - \epsilon/\|\tilde{\mu}\|_2) - q/2|}{\|\tilde{\mu}(1 - \epsilon/\|\tilde{\mu}\|_2)\|_2} \frac{2\|\tilde{\mu}(1 - \epsilon/\|\tilde{\mu}\|_2)\|_2}{|\tilde{\mu}^T\tilde{\mu}(1 - \epsilon/\|\tilde{\mu}\|_2) - q/2| + |\tilde{\mu}^T\tilde{\mu}(1 - \epsilon/\|\tilde{\mu}\|_2) + q/2|}$$

$$= \frac{2|(x - \frac{\mu_1+\mu_2}{2})^T \tilde{\mu}(1 - \epsilon/\|\tilde{\mu}\|_2) - q/2|}{|\tilde{\mu}^T\tilde{\mu}(1 - \epsilon/\|\tilde{\mu}\|_2) - q/2| + |\tilde{\mu}^T\tilde{\mu}(1 - \epsilon/\|\tilde{\mu}\|_2) + q/2|}.$$

(iii) For sample $x \sim P_{\mu_1,\mu_2,I_d}$, consider the Bayes optimal robust classifier in equation 4, we can calculate the analytical standard accuracy by

$$\mathbb{P}(y = 1)\mathbb{P}\left[f_\epsilon(x) = 1 \mid y = 1\right] + \mathbb{P}(y = -1)\mathbb{P}\left[f_\epsilon(x) = -1 \mid y = -1\right]$$

$$= \tau\mathbb{P}\left[f_\epsilon(x) = 1 \mid y = 1\right] + (1 - \tau)\left[f_\epsilon(x) = -1 \mid y = -1\right]$$

$$= \tau\mathbb{P}\left[(x - \frac{\mu_1 + \mu_2}{2})^T \tilde{\mu}(1 - \epsilon/\|\tilde{\mu}\|_2) - q/2 > 0 \mid y = 1\right]$$

$$+ (1 - \tau)\mathbb{P}\left[(x - \frac{\mu_1 + \mu_2}{2})^T \tilde{\mu}(1 - \epsilon/\|\tilde{\mu}\|_2) - q/2 < 0 \mid y = -1\right]$$

$$= \tau\mathbb{P}\left[(\tilde{\mu} + w)^T \tilde{\mu}(1 - \epsilon/\|\tilde{\mu}\|_2) - q/2 > 0\right],$$

$$+ (1 - \tau)\mathbb{P}\left[(-\tilde{\mu} + w)^T \tilde{\mu}(1 - \epsilon/\|\tilde{\mu}\|_2) - q/2 < 0\right], \quad w \sim \mathcal{N}(0, I_d)$$

$$= \tau\mathbb{P}\left[w^T \tilde{\mu}(1 - \epsilon/\|\tilde{\mu}\|_2) > q/2 - \tilde{\mu}^T \tilde{\mu}(1 - \epsilon/\|\tilde{\mu}\|_2)\right],$$

$$+ (1 - \tau)\mathbb{P}\left[w^T \tilde{\mu}(1 - \epsilon/\|\tilde{\mu}\|_2) < q/2 + \tilde{\mu}^T \tilde{\mu}(1 - \epsilon/\|\tilde{\mu}\|_2)\right], \quad w \sim \mathcal{N}(0, I_d)$$

$$= \tau\mathbb{P}\left[\frac{w^T \tilde{\mu}(1 - \epsilon/\|\tilde{\mu}\|_2)}{\|\tilde{\mu}(1 - \epsilon/\|\tilde{\mu}\|_2)\|_2} > \frac{q/2 - \tilde{\mu}^T \tilde{\mu}(1 - \epsilon/\|\tilde{\mu}\|_2)}{\|\tilde{\mu}(1 - \epsilon/\|\tilde{\mu}\|_2)\|_2}\right],$$

$$+ (1 - \tau)\mathbb{P}\left[\frac{w^T \tilde{\mu}(1 - \epsilon/\|\tilde{\mu}\|_2)}{\|\tilde{\mu}(1 - \epsilon/\|\tilde{\mu}\|_2)\|_2} < \frac{q/2 + \tilde{\mu}^T \tilde{\mu}(1 - \epsilon/\|\tilde{\mu}\|_2)}{\|\tilde{\mu}(1 - \epsilon/\|\tilde{\mu}\|_2)\|_2}\right], \quad \frac{w^T \tilde{\mu}(1 - \epsilon/\|\tilde{\mu}\|_2)}{\|\tilde{\mu}(1 - \epsilon/\|\tilde{\mu}\|_2)\|_2} \sim \mathcal{N}(0, 1)$$

$$= \tau\Phi(\frac{\tilde{\mu}^T \tilde{\mu}(1 - \epsilon/\|\tilde{\mu}\|_2) - q/2}{\|\tilde{\mu}(1 - \epsilon/\|\tilde{\mu}\|_2)\|_2}) + (1 - \tau)\Phi(\frac{\tilde{\mu}^T \tilde{\mu}(1 - \epsilon/\|\tilde{\mu}\|_2) + q/2}{\|\tilde{\mu}(1 - \epsilon/\|\tilde{\mu}\|_2)\|_2}).$$

Let $w$ denote $\tilde{\mu}(1 - \epsilon/\|\tilde{\mu}\|_2)$, the we got the accuracy

$$a = \tau\Phi(\frac{\tilde{\mu}^T w - q/2}{\|w\|_2}) + (1 - \tau)\Phi(\frac{\tilde{\mu}^T w + q/2}{\|w\|_2}).$$

(iv) For sample $x \sim P_{\mu_1,\mu_2,I_d}$, let $t$ denote $x - \frac{\mu_1+\mu_2}{2}$, and $w$ denote $\tilde{\mu}(1 - \epsilon/\|\tilde{\mu}\|_2)$. According to Theorem 4(iii), when $\tilde{\mu}^T \tilde{\mu}(1 - \epsilon/\|\tilde{\mu}\|_2) - q/2 > 0$, the accuracy would be higher than 0.5. Therefore we consider $\tilde{\mu}^T w - q/2 > 0$.

Now consider the classifier in equation 5 and the corresponding scaled bound from (ii),

$$\|\bar{\Delta}\|_2 = \frac{2|(x - \frac{\mu_1+\mu_2}{2})^T \tilde{\mu}(1 - \epsilon/\|\tilde{\mu}\|_2) - q/2|}{|\tilde{\mu}^T\tilde{\mu}(1 - \epsilon/\|\tilde{\mu}\|_2) - q/2| + |\tilde{\mu}^T\tilde{\mu}(1 - \epsilon/\|\tilde{\mu}\|_2) + q/2|} = \frac{2|t^T w - q/2|}{|\tilde{\mu}^T w - q/2| + |\tilde{\mu}^T w + q/2|}.$$

Since $t|y \sim \mathcal{N}(y\tilde{\mu}, I_d)$, we have $t^T w - q/2|y \sim \mathcal{N}(y\tilde{\mu}^T w - q/2, w^T w)$. When we only want to get the expected scaled bound of the correctly-classified samples, we have that

$$\mathbb{E}\left[\|\bar{\Delta}\|_2 \mid f_\epsilon(x) = y\right]$$

$$= \frac{2}{|\tilde{\mu}^T w - q/2| + |\tilde{\mu}^T w + q/2|} \mathbb{E}\left[|t^T w - q/2| \mid f_\epsilon(x) = y\right]$$

$$= \frac{\tau\Phi(\frac{\tilde{\mu}^T w - q/2}{\|w\|_2})}{\tau\Phi(\frac{\tilde{\mu}^T w - q/2}{\|w\|_2}) + (1-\tau)\Phi(\frac{\tilde{\mu}^T w + q/2}{\|w\|_2})} \frac{2}{|\tilde{\mu}^T w - q/2| + |\tilde{\mu}^T w + q/2|} \mathbb{E}\left[|t^T w - q/2| \mid f_\epsilon(x) = y = 1\right]$$

$$+ \frac{(1-\tau)\Phi(\frac{\tilde{\mu}^T w + q/2}{\|w\|_2})}{\tau\Phi(\frac{\tilde{\mu}^T w - q/2}{\|w\|_2}) + (1-\tau)\Phi(\frac{\tilde{\mu}^T w + q/2}{\|w\|_2})} \frac{2}{|\tilde{\mu}^T w - q/2| + |\tilde{\mu}^T w + q/2|} \mathbb{E}\left[|t^T w - q/2| \mid f_\epsilon(x) = y = -1\right]$$

$$= \frac{\tau\Phi(\frac{\tilde{\mu}^T w - q/2}{\|w\|_2})}{\tau\Phi(\frac{\tilde{\mu}^T w - q/2}{\|w\|_2}) + (1-\tau)\Phi(\frac{\tilde{\mu}^T w + q/2}{\|w\|_2})} \frac{2}{|\tilde{\mu}^T w - q/2| + |\tilde{\mu}^T w + q/2|} \mathbb{E}\left[t^T w - q/2 \mid y = 1, t^T w - q/2 \geq 0\right]$$

$$+ \frac{(1-\tau)\Phi(\frac{\tilde{\mu}^T w + q/2}{\|w\|_2})}{\tau\Phi(\frac{\tilde{\mu}^T w - q/2}{\|w\|_2}) + (1-\tau)\Phi(\frac{\tilde{\mu}^T w + q/2}{\|w\|_2})} \frac{2}{|\tilde{\mu}^T w - q/2| + |\tilde{\mu}^T w + q/2|} \mathbb{E}\left[-t^T w + q/2 \mid y = -1, t^T w - q/2 < 0\right].$$

Recall that $t^T w - q/2|y \sim \mathcal{N}(y\tilde{\mu}^T w - q/2, w^T w)$, then by the mean of truncated normal distribution, it is true that

$$\mathbb{E}\left[t^T w - q/2 \mid y = 1, t^T w - q/2 \geq 0\right] = \tilde{\mu}^T w - q/2 + \|w\|_2 \frac{\phi(\frac{0 - \tilde{\mu}^T w + q/2}{\|w\|_2})}{1 - \Phi(\frac{0 - \tilde{\mu}^T w + q/2}{\|w\|_2})}$$

$$= \tilde{\mu}^T w - q/2 + \|w\|_2 \frac{\phi(\frac{-\tilde{\mu}^T w + q/2}{\|w\|_2})}{\Phi(\frac{\tilde{\mu}^T w - q/2}{\|w\|_2})}$$

$$\mathbb{E}\left[-t^T w + q/2 \mid y = -1, t^T w - q/2 < 0\right] = -\mathbb{E}\left[t^T w - q/2 \mid y = -1, t^T w - q/2 < 0\right]$$

$$= -\left(-\tilde{\mu}^T w - q/2 - \|w\|_2 \frac{\phi(\frac{0 + \tilde{\mu}^T w + q/2}{\|w\|_2})}{\Phi(\frac{0 + \tilde{\mu}^T w + q/2}{\|w\|_2})}\right)$$

$$= \tilde{\mu}^T w + q/2 + \|w\|_2 \frac{\phi(\frac{\tilde{\mu}^T w + q/2}{\|w\|_2})}{\Phi(\frac{\tilde{\mu}^T w + q/2}{\|w\|_2})}$$

Therefore

$$\mathbb{E}\left[\|\bar{\Delta}\|_2 \mid f_\epsilon(x) = y\right]$$

$$= \frac{\tau\Phi(\frac{\tilde{\mu}^T w - q/2}{\|w\|_2})}{\tau\Phi(\frac{\tilde{\mu}^T w - q/2}{\|w\|_2}) + (1-\tau)\Phi(\frac{\tilde{\mu}^T w + q/2}{\|w\|_2})} \frac{2}{|\tilde{\mu}^T w - q/2| + |\tilde{\mu}^T w + q/2|} \left(\tilde{\mu}^T w - q/2 + \|w\|_2 \frac{\phi(\frac{-\tilde{\mu}^T w + q/2}{\|w\|_2})}{\Phi(\frac{\tilde{\mu}^T w - q/2}{\|w\|_2})}\right)$$

$$+ \frac{(1-\tau)\Phi(\frac{\tilde{\mu}^T w + q/2}{\|w\|_2})}{\tau\Phi(\frac{\tilde{\mu}^T w - q/2}{\|w\|_2}) + (1-\tau)\Phi(\frac{\tilde{\mu}^T w + q/2}{\|w\|_2})} \frac{2}{|\tilde{\mu}^T w - q/2| + |\tilde{\mu}^T w + q/2|} \left(\tilde{\mu}^T w + q/2 + \|w\|_2 \frac{\phi(\frac{\tilde{\mu}^T w + q/2}{\|w\|_2})}{\Phi(\frac{\tilde{\mu}^T w + q/2}{\|w\|_2})}\right)$$

$$\qquad\qquad\qquad\qquad\qquad\qquad\qquad\qquad\qquad\qquad\qquad\qquad\qquad\qquad\qquad\qquad\qquad\qquad\qquad\qquad\qquad\qquad\qquad \square$$

A.6 ALGORITHM

---

**Algorithm 1** Evaluating Pretrained Image Representations using Synthetic Data (*SynBench*)

---

**Input**: A representation network $g_\theta : \mathbb{R}^d \to \mathbb{R}^{d'}$, threshold accuracy $a_T$, (optional) the probability density function of the synthetic data manifold $\mathbb{P}_\mu$ and $\mathbb{P}_\Sigma$.

**Output**: SynBench-score that quantifies the robustness-accuracy performance.

1: **if** $\mathbb{P}_\mu$ and $\mathbb{P}_\Sigma$ are specified **then**
2:     $\mu \sim \mathbb{P}_\mu, \Sigma \sim \mathbb{P}_\Sigma$.
3: **else**
4:     $\mu = s \cdot 1_d / \sqrt{d}, \ s \sim \mathcal{U}\{0.1, 5\}$, and $\Sigma = I_d$.
5: **end if**
6: Draw $n$ synthetic data hyper-parameters $\{(\mu_k, \Sigma_k)\}_{k=1}^n$.
7: **for** $k \leftarrow 1$ to $n$ **do**
8:     Generate class-conditional Gaussian data $(x^{train}, y^{train})$ and test set $(x^{test}, y^{test})$ following $x - \bar{\mu}|y \sim \mathcal{N}(y\mu_k, \Sigma_k)$ and $\bar{\mu} = 0.5 \cdot 1_d / \sqrt{d}$.
9:     Calculate $a_k^{\text{input}}$, the theoretical accuracy for input data, following Thm 1(iii).
10:     Calculate $b_k^{\text{input}}$ (denotes $\mathbb{E}\left[\|\bar{\Delta}_x\|_2 \mid f_\epsilon(x) = y\right]$), the expected scaled bound of correct samples for input data, following Thm 1(iv).
11:     Gather representations for class 1 training samples $z_1^{train,i} = g_\theta(x^{train,i})$ if $y^{train,i} = 1$, representations for class 2 training samples $z_2^{train,j} = g_\theta(x^{train,j})$ if $y^{train,j} = -1$, and $z^{test} = g_\theta(x^{test})$.
12:     Estimate class-conditional Gaussian in the representation space by $\mu_1' = \frac{\sum_{i=1}^{n_1} z_1^{train,i}}{n_1}$, $\mu_2' = \frac{\sum_{j=1}^{n_2} z_2^{train,j}}{n_2}$, $\Sigma' = \frac{\sum_{i=1}^{n_1}(z_1^{train,i}-\mu_1')(z_1^{train,i}-\mu_1')^T + \sum_{j=1}^{n_2}(z_2^{train,j}-\mu_2')(z_2^{train,j}-\mu_2')^T}{n_1+n_2-1}$.
13:     Derive Bayes optimal classifier $f_\epsilon'$ for class-conditional Gaussian distribution $z|y = 1 \sim \mathcal{N}(\mu_1', \Sigma'), z|y = -1 \sim \mathcal{N}(\mu_2', \Sigma')$.
14:     Calculate $a_k^{\text{repre}}$, the accuracy of $f_\epsilon'$ for representations $z^{test}$, empirically.
15:     Calculate the scaled bound of correct samples for representations following Thm 1(ii), $\|\bar{\Delta}_z\|_2 = \frac{|(z^{test} - \frac{\mu_1'+\mu_2'}{2})^T \Sigma'^{-1}(\tilde{\mu}-z_{\Sigma'}(\tilde{\mu}))|}{|\tilde{\mu}^T \Sigma'^{-1}(\tilde{\mu}-z_{\Sigma'}(\tilde{\mu}))|}$ where $\tilde{\mu} = \frac{\mu_1'-\mu_2'}{2}$.
16:     Estimate $b_k^{\text{repre}}$, the expected scaled bound of correct samples for representations empirically, by the arithmetic mean.
17: **end for**
18: Calculate $E(a_t)$ for input data with $\{a_k^{\text{input}}, b_k^{\text{input}}\}_{k=1}^n$ according to equation 2.
19: Calculate $E_{\theta,\epsilon}(a_t)$ for representations with $\{a_k^{\text{repre}}, b_k^{\text{repre}}\}_{k=1}^n$ according to equation 2.
20: Calculate SynBench-Score$(\theta, \epsilon, a_T) = \frac{\int_{a_T}^1 E_{\theta,\epsilon}(a_t) da_t}{\int_{a_T}^1 E(a_t) da_t}$.

---

## A.7 MODEL DESCRIPTIONS

We list and compare 10 pretrained vision transformers (ViTs)[1](Dosovitskiy et al., 2020; Chen et al., 2021; Caron et al., 2021) and ResNets[2](Chen et al., 2020c) in the following table.

| Model | Arch. | pretraining | fine-tuning | patch | # parameters (M) |
|---|---|---|---|---|---|
| ViT-Ti/16 | ViT-Tiny | Imgn21k | Imgn1k | 16 | 5.7 |
| ViT-B/16 | ViT-Base | Imgn21k | Imgn1k | 16 | 86.6 |
| ViT-B/16-in21k | ViT-Base | Imgn21k | No | 16 | 86.6 |
| ViT-L/16 | ViT-Large | Imgn21k | Imgn1k | 16 | 304.3 |
| ViT-S/16-DINO | ViT-Small | self-Imgn1k | No | 16 | 21.7 |
| ViT-S/8-DINO | ViT-Small | self-Imgn1k | No | 8 | 21.7 |
| ViT-B/16-DINO | ViT-Base | self-Imgn1k | No | 16 | 85.8 |
| ViT-B/8-DINO | ViT-Base | self-Imgn1k | No | 8 | 85.8 |
| Resnet50-SimCLRv2 | Resnet50 | self-Imgn1k | No | - | 144.4 |
| Resnet101-SimCLRv2 | Resnet101 | self-Imgn1k | No | - | 261.2 |
| Variation: | | | | | |
| Model size | ViT-{Ti,B,L}/16, ViT-{S,B}/16-DINO, ViT-{S,B}/8-DINO, Resnet{50,101}-SimCLRv2 | | | | |
| Finetuning | ViT-B/16{,-in21k} | | | | |
| ViT patch size | ViT-S/{16,8}-DINO, ViT-B/{16,8}-DINO | | | | |

Table 5: Model descriptions. The performance of models might be nuanced by scheduler, curriculum, and training episodes, which are not captured in the table.

## A.8 RUNTIME ANALYSIS

The runtime of SynBench depends on the number of outcomes of the discrete uniform distribution $\mathcal{U}\{0.1, 5\}$ and the data inference time through the pretrained model. For one outcome (one robustness-accuracy relationship), it costs 59 seconds to generate 2048 Gaussian samples, 37 and 81 seconds to obtain the SynBench-Score for ViT-B/16 and ViT-L/16 on one GeForce RTX 2080 super.

Correspondingly, to obtain one robustness-accuracy relationship with task-specific methods requires us to perform adversarial attacks on multiple possible datasets. Here, we ignore to the time to train the linear probing layer. For one single dataset, e.g. CIFAR10, AutoAttack uses 72320 and 332288 seconds to evaluate 2048 samples on ViT-B/16 and ViT-L/16 on one GeForce RTX 2080 super; PGD attack uses 1280 and 4608 seconds to evaluate 2048 samples on ViT-B/16 and ViT-L/16 on one GeForce RTX 2080 super.

For other task-agnostic metrics (MDL, SDL, $\epsilon$SC), obtaining them for ViT-B/16 costs 6807 seconds and ViT-L/16 costs 7373 seconds on one Tesla V100. However, it should be noted that these metrics do not indicate robustness performance.

---

[1]https://github.com/rwightman/pytorch-image-models
[2]https://github.com/google-research/simclr

## A.9 ADDITIONAL TABLES

| $a_t$ | Model | $\epsilon=0$ | $\epsilon=0.1$ | $\epsilon=0.2$ | $\epsilon=0.3$ | $\epsilon=0.4$ | $\epsilon=0.5$ | $\epsilon=0.6$ | $\epsilon=0.7$ | $\epsilon=0.8$ |
|---|---|---|---|---|---|---|---|---|---|---|
| 0.7 | ViT-Ti/16 | 0.01 | 0.01 | 0 | 0 | 0 | 0 | 0 | 0 | 0 |
| | ViT-B/16 | 0.33 | 0.36 | 0.37 | 0.35 | 0.32 | 0.27 | 0.20 | 0.13 | 0.07 |
| | ViT-B/16-in21k | 0.20 | 0.22 | 0.23 | 0.21 | 0.17 | 0.13 | 0.07 | 0.03 | 0.01 |
| | ViT-L/16 | 0.26 | 0.30 | 0.33 | 0.32 | 0.30 | 0.27 | 0.22 | 0.17 | 0.11 |
| | ViT-S/16-DINO | 0.48 | 0.48 | 0.47 | 0.45 | 0.42 | 0.37 | 0.32 | 0.25 | 0.17 |
| | ViT-B/16-DINO | 0.55 | 0.58 | 0.58 | 0.56 | 0.53 | 0.50 | 0.46 | 0.41 | 0.35 |
| | ViT-S/8-DINO | 0.40 | 0.42 | 0.42 | 0.41 | 0.39 | 0.37 | 0.34 | 0.30 | 0.26 |
| | ViT-B/8-DINO | 0.50 | 0.55 | 0.56 | 0.54 | 0.50 | 0.45 | 0.40 | 0.35 | 0.30 |
| | Resnet50-SimCLRv2 | 0.66 | 0.53 | 0.50 | 0.49 | 0.50 | 0.49 | 0.48 | 0.48 | 0.48 |
| | Resnet101-SimCLRv2 | 0.60 | 0.74 | 0.64 | 0.58 | 0.56 | 0.52 | 0.51 | 0.50 | 0.48 |
| 0.75 | ViT-Ti/16 | 0 | 0 | 0 | 0 | 0 | 0 | 0 | 0 | 0 |
| | ViT-B/16 | 0.26 | 0.29 | 0.30 | 0.28 | 0.25 | 0.18 | 0.11 | 0.05 | 0.01 |
| | ViT-B/16-in21k | 0.12 | 0.15 | 0.16 | 0.14 | 0.10 | 0.06 | 0.02 | 0 | 0 |
| | ViT-L/16 | 0.19 | 0.24 | 0.27 | 0.27 | 0.24 | 0.21 | 0.16 | 0.10 | 0.04 |
| | ViT-S/16-DINO | 0.42 | 0.42 | 0.41 | 0.39 | 0.36 | 0.32 | 0.26 | 0.19 | 0.11 |
| | ViT-B/16-DINO | 0.50 | 0.54 | 0.54 | 0.51 | 0.48 | 0.44 | 0.39 | 0.34 | 0.27 |
| | ViT-S/8-DINO | 0.33 | 0.34 | 0.34 | 0.33 | 0.31 | 0.29 | 0.25 | 0.21 | 0.16 |
| | ViT-B/8-DINO | 0.44 | 0.50 | 0.51 | 0.48 | 0.43 | 0.37 | 0.31 | 0.25 | 0.19 |
| | Resnet50-SimCLRv2 | 0.33 | 0.44 | 0.40 | 0.39 | 0.39 | 0.39 | 0.39 | 0.39 | 0.39 |
| | Resnet101-SimCLRv2 | 0.54 | 0.69 | 0.57 | 0.49 | 0.45 | 0.43 | 0.40 | 0.38 | 0.36 |
| 0.8 | ViT-Ti/16 | 0 | 0 | 0 | 0 | 0 | 0 | 0 | 0 | 0 |
| | ViT-B/16 | 0.19 | 0.22 | 0.23 | 0.21 | 0.17 | 0.11 | 0.04 | 0 | 0 |
| | ViT-B/16-in21k | 0.06 | 0.08 | 0.09 | 0.07 | 0.04 | 0.01 | 0 | 0 | 0 |
| | ViT-L/16 | 0.12 | 0.17 | 0.21 | 0.20 | 0.18 | 0.14 | 0.09 | 0.04 | 0 |
| | ViT-S/16-DINO | 0.34 | 0.35 | 0.34 | 0.32 | 0.29 | 0.25 | 0.19 | 0.13 | 0.05 |
| | ViT-B/16-DINO | 0.45 | 0.49 | 0.49 | 0.46 | 0.42 | 0.37 | 0.32 | 0.26 | 0.17 |
| | ViT-S/8-DINO | 0.25 | 0.26 | 0.26 | 0.25 | 0.23 | 0.20 | 0.16 | 0.12 | 0.08 |
| | ViT-B/8-DINO | 0.38 | 0.45 | 0.46 | 0.42 | 0.36 | 0.29 | 0.22 | 0.16 | 0.10 |
| | Resnet50-SimCLRv2 | 0.09 | 0.34 | 0.31 | 0.31 | 0.30 | 0.30 | 0.30 | 0.30 | 0.30 |
| | Resnet101-SimCLRv2 | 0.46 | 0.62 | 0.50 | 0.39 | 0.35 | 0.32 | 0.29 | 0.26 | 0.24 |
| 0.85 | ViT-Ti/16 | 0 | 0 | 0 | 0 | 0 | 0 | 0 | 0 | 0 |
| | ViT-B/16 | 0.10 | 0.14 | 0.15 | 0.13 | 0.09 | 0.04 | 0 | 0 | 0 |
| | ViT-B/16-in21k | 0.01 | 0.03 | 0.02 | 0.02 | 0 | 0 | 0 | 0 | 0 |
| | ViT-L/16 | 0.05 | 0.09 | 0.13 | 0.12 | 0.10 | 0.07 | 0.03 | 0 | 0 |
| | ViT-S/16-DINO | 0.25 | 0.26 | 0.25 | 0.24 | 0.21 | 0.17 | 0.12 | 0.06 | 0.01 |
| | ViT-B/16-DINO | 0.38 | 0.43 | 0.43 | 0.40 | 0.35 | 0.29 | 0.23 | 0.15 | 0.08 |
| | ViT-S/8-DINO | 0.16 | 0.17 | 0.17 | 0.15 | 0.13 | 0.10 | 0.07 | 0.04 | 0.02 |
| | ViT-B/8-DINO | 0.30 | 0.38 | 0.38 | 0.34 | 0.27 | 0.19 | 0.13 | 0.06 | 0.03 |
| | Resnet50-SimCLRv2 | 0 | 0.22 | 0.20 | 0.20 | 0.19 | 0.19 | 0.19 | 0.20 | 0.19 |
| | Resnet101-SimCLRv2 | 0.37 | 0.55 | 0.41 | 0.28 | 0.23 | 0.19 | 0.16 | 0.13 | 0.11 |
| 0.9 | ViT-Ti/16 | 0 | 0 | 0 | 0 | 0 | 0 | 0 | 0 | 0 |
| | ViT-B/16 | 0.02 | 0.04 | 0.05 | 0.04 | 0.01 | 0 | 0 | 0 | 0 |
| | ViT-B/16-in21k | 0 | 0 | 0 | 0 | 0 | 0 | 0 | 0 | 0 |
| | ViT-L/16 | 0 | 0.01 | 0.04 | 0.04 | 0.03 | 0.01 | 0 | 0 | 0 |
| | ViT-S/16-DINO | 0.13 | 0.14 | 0.14 | 0.13 | 0.10 | 0.07 | 0.04 | 0.01 | 0 |
| | ViT-B/16-DINO | 0.28 | 0.34 | 0.34 | 0.30 | 0.23 | 0.16 | 0.10 | 0.05 | 0 |
| | ViT-S/8-DINO | 0.05 | 0.06 | 0.06 | 0.05 | 0.04 | 0.02 | 0.01 | 0 | 0 |
| | ViT-B/8-DINO | 0.20 | 0.29 | 0.28 | 0.23 | 0.15 | 0.07 | 0.03 | 0 | 0 |
| | Resnet50-SimCLRv2 | 0 | 0.08 | 0.06 | 0.06 | 0.06 | 0.06 | 0.06 | 0.06 | 0.06 |
| | Resnet101-SimCLRv2 | 0.23 | 0.42 | 0.28 | 0.15 | 0.08 | 0.06 | 0.04 | 0.02 | 0.01 |

Table 6: Full table of Table 1.

## A.10 OTHER BASELINES

For completeness, we report several baseline metrics for the synthetic conditional Gaussian classification task. We follow the implementation of Whitney et al. (2020); Shao et al. (2022) and set the training set size $n$ to be $2048, 8192, 32768$. In Table 7, we report validation loss (val loss), minimum description length (MDL) (Voita & Titov, 2020), surplus description length (SDL), $\epsilon$-sample complexity ($\epsilon$-SC) (Whitney et al., 2020), logarithm of maximum evidence (LogME) (You et al., 2021) and self-challenging Fisher discriminant analysis (SFDA) (Shao et al., 2022) on our synthetic proxy task as baselines. We aim at calculating the Pearson correlation between task-agnostic metrics and possible downstream tasks. We take the average accuracy of 27 downstream tasks in the literature (Radford et al., 2021) for each pretrained model and treat it as the real-life performance measure. For an even more complete picture, we also consider some synthetic distribution shifts that include image corruptions (ImageNet-c), style transfer (ImageNet-r), and adversarial examples (ImageNet-a). To analyze how data with these synthetic distribution shifts can inform general pretrained models'

performance, we quoted the their accuracy from Wightman (2019) and calculated their correlation with the average real-life accuracy in Table 7. Furthermore, following Zhang et al. (2021), we perform "partially corrupted labels" experiments on CIFAR10 dataset with the level of label corruptions equals to 0.5. See line "CIFAR10-lc acc." for the results. We note that the correlation coefficients in these four cases suggest only moderate correlation to even negative correlation.

We set the training set size $n$ to be $2048, 4096, 8192, 16384, 32768$ and compare the model selections between ViT-B/16 and ViT-B/16-in21k in Table 8. In Table 9, we report these metrics on all 10 pretrained representations for $n = 8192$.

| $n$ | Name | ViT-B/16 | ViT-L/16 | ViT-B/32 | Resnet50-SimCLRv2 | Resnet101-SimCLRv2 | Pearson correlation |
|---|---|---|---|---|---|---|---|
| Reallife | Accuracy (%) | 74.3 | 75.5 | 72.6 | 75.4 | 75.4 | 1.0 |
| Transfer dataset | ImageNet-c acc. | 66.4 | 72.2 | 61.4 | 47.4 | 50.1 | 0.64 |
| | ImageNet-r acc. | 56.8 | 64.3 | 49.4 | 39.4 | 44.1 | -0.03 |
| | ImageNet-a acc. | 43.1 | 55.3 | 22.3 | 27.1 | 38.2 | 0.57 |
| | CIFAR10-lc acc. | 93.54 | 94.95 | 92.48 | 85.74 | 87.38 | -0.36 |
| 2048 | Val loss | 3.10 | 4.12 | 4.10 | 1.31 | 0.98 | -0.55 |
| | MDL | 6820.76 | 8094.06 | 8198.55 | 5881.34 | 2882.36 | -0.50 |
| | SDL, $\varepsilon = 1$ | $> 4977.76$ | $> 6251.06$ | $> 6355.55$ | $> 4038.34$ | 1052.37 | - |
| | $\varepsilon$SC, $\varepsilon = 1$ | $> 1843.0$ | $> 1843.0$ | $> 1843.0$ | $> 1843.0$ | 1843 | - |
| | LogME | -0.726 | -0.724 | -0.729 | 2.791 | 1.503 | 0.54 |
| | SFDA | 0.584 | 0.635 | 0.567 | 0.947 | 0.593 | 0.46 |
| | SynBench | 0.33 | 0.26 | 0.02 | 0.66 | 0.60 | **0.79** |
| 8192 | Val loss | 0.73 | 1.50 | 2.92 | 0.62 | 0.52 | -0.81 |
| | MDL | 9939.13 | 17672.6 | 23332.98 | 9646.09 | 5443.43 | -0.68 |
| | SDL, $\varepsilon = 1$ | 3479.59 | $> 10300.6$ | $> 15960.98$ | 3700.73 | 776.38 | - |
| | $\varepsilon$SC, $\varepsilon = 1$ | 7372 | $> 7372.0$ | $> 7372.0$ | 4045 | 669 | - |
| | LogME | -0.710 | -0.707 | -0.727 | -0.599 | -0.622 | 0.65 |
| | SFDA | 0.525 | 0.531 | 0.513 | 0.581 | 0.543 | 0.67 |
| | SynBench | 0.52 | 0.49 | 0.01 | 0.69 | 0.84 | **0.89** |
| 32768 | Val loss | 0.68 | 0.79 | 3.91 | 0.53 | 0.51 | -0.92 |
| | MDL | 30848.99 | 38718.04 | 107960.49 | 22022.08 | 17166.0 | -0.91 |
| | SDL, $\varepsilon = 1$ | 7043.32 | 12496.0 | $> 78469.49$ | 4355.67 | 969.27 | - |
| | $\varepsilon$SC, $\varepsilon = 1$ | 14265 | 29491 | $> 29491.0$ | 3338 | 1615 | - |
| | LogME | -0.686 | -0.687 | -0.725 | -0.580 | -0.608 | 0.72 |
| | SFDA | 0.517 | 0.518 | 0.505 | 0.545 | 0.534 | 0.77 |
| | SynBench | 0.59 | 0.58 | 0.02 | 0.81 | 0.87 | **0.92** |

Table 7: Pearson correlation between task agnostic metrics and the average accuracy on 27 real-life tasks (Radford et al., 2021, Table 10) . We report the 5 pretrained models out of the overall 10 due to the lack of reported results from the literature for the other pretrain models.

| n | Name | ViT-B/16 | ViT-B/16-in21k |
|---|---|---|---|
| 2048 | Val loss | 3.10 | 3.37 |
| | MDL | 6820.76 | 7114.12 |
| | SDL, $\varepsilon$=1 | $> 4977.76$ | $> 5271.12$ |
| | $\varepsilon$SC, $\varepsilon$=1 | $> 1843.0$ | $> 1843.0$ |
| | SynBench | 0.33 | 0.20 |
| 4096 | Val loss | 1.77 | 1.41 |
| | MDL | 10813.95 | 9412.53 |
| | SDL, $\varepsilon$=1 | $> 7127.95$ | $> 5726.53$ |
| | $\varepsilon$SC, $\varepsilon$=1 | $> 3686.0$ | $> 3686.0$ |
| | SynBench | 0.45 | 0.30 |
| 8192 | Val loss | 0.73 | 0.77 |
| | MDL | 9939.13 | 9773.16 |
| | SDL, $\varepsilon$=1 | 3479.59 | 3153.33 |
| | $\varepsilon$SC, $\varepsilon$=1 | 7372 | 7372 |
| | SynBench | 0.52 | 0.38 |
| 16384 | Val loss | 0.85 | 0.86 |
| | MDL | 20936.18 | 20899.58 |
| | SDL, $\varepsilon$=1 | 7266.8 | 7136.29 |
| | $\varepsilon$SC, $\varepsilon$=1 | 14745 | 14745 |
| | SynBench | 0.56 | 0.41 |
| 32768 | Val loss | 0.68 | 0.70 |
| | MDL | 30848.99 | 32944.76 |
| | SDL, $\varepsilon$=1 | 7043.32 | 8611.49 |
| | $\varepsilon$SC, $\varepsilon$=1 | 14265 | 14265 |
| | SynBench | 0.59 | 0.44 |

Table 8: Baseline metrics evaluating the representation quality on the conditional Gaussian synthetic data with $n = \{2048, 4096, 8192, 16384, 32768\}$. For Val loss, MDL, SDL, and $\epsilon$SC, the smaller the better; for SynBench, the bigger the better. Note that the model ranking of SynBench is consistent across different values of $n$, while other methods will change their rankings.

| Name | Val loss | MDL | SDL, $\varepsilon$=1 | $\varepsilon$SC, $\varepsilon$=1 |
|---|---|---|---|---|
| ViT-Ti/16 | 4.38 | 30071.64 | > 22699.64 | > 7372.0 |
| ViT-B/16 | 0.73 | 9939.13 | 3479.59 | 7372 |
| ViT-L/16 | 1.50 | 17672.6 | > 10300.6 | > 7372.0 |
| ViT-B/16-in21k | 0.77 | 9773.16 | 3153.33 | 7372 |
| ViT-S/16-DINO | 1.51 | 18536.93 | > 11164.93 | > 7372.0 |
| ViT-S/8-DINO | 0.70 | 8196.8 | 2056.69 | 4045 |
| ViT-B/16-DINO | 0.92 | 10535.11 | 3432.28 | 7372 |
| ViT-B/8-DINO | 0.64 | 6796.87 | 1185.31 | 2220 |
| Resnet50-SimCLRv2 | 0.62 | 9646.09 | 3700.73 | 4045 |
| Resnet101-SimCLRv2 | 0.52 | 5443.43 | 776.38 | 669 |

Table 9: Baseline metrics evaluating the representation quality on the conditional Gaussian synthetic data with $n = 8192$.

## A.11 SYNTHETIC DATASET WITH MODELED COVARIANCE

| Models | CIFAR10 | | SVHN | |
|---|---|---|---|---|
| | SA | RA | SA | RA |
| ViT-Ti/16 | 81.9 | 1.1 | 48.0 | 0.7 |
| ViT-B/16 | 95.0 | 32.1 | 65.4 | 5.2 |
| ViT-B/16-in21k | 88.3 | 15.7 | 64.7 | 3.2 |
| ViT-L/16 | 98.0 | 57.0 | 68.9 | 8.4 |
| ViT-S/16-DINO | 95.3 | 0 | 70.2 | 0 |
| ViT-B/16-DINO | 96.5 | 4.7 | 72.7 | 1.0 |
| ViT-S/8-DINO | 96.2 | 0 | 73.0 | 0 |
| ViT-B/8-DINO | 97.0 | 0 | 74.2 | 0 |
| Res50-SimCLRv2 | 95.0 | 0 | 74.2 | 0 |
| Res101-SimCLRv2 | 95.6 | 0 | 71.7 | 0 |

Table 10: Full Table of Table 3.

| Dataset | Distance | Gaussian-I | Gaussian-H |
|---|---|---|---|
| CIFAR10 | FID | 466 | 454 |
| | MD | 1583 | 1483 |
| SVHN | FID | 503 | 494 |
| | MD | 1372 | 1237 |
| TinyImageNet | FID | 521 | 494 |
| | MD | 1636 | 1320 |

Table 11: Distances from synthetic data to CIFAR10, SVHN, and TinyImageNet.

| $a_t = 0.7$ | $\epsilon = 0$ | $\epsilon = 0.1$ | $\epsilon = 0.2$ | $\epsilon = 0.3$ | $\epsilon = 0.4$ | $\epsilon = 0.5$ | $\epsilon = 0.6$ | $\epsilon = 0.7$ | $\epsilon = 0.8$ |
|---|---|---|---|---|---|---|---|---|---|
| ViT-B/16 | 0.18 | 0.22 | 0.24 | 0.23 | 0.20 | 0.15 | 0.10 | 0.05 | 0.01 |
| ViT-B/16-in21k | 0.07 | 0.10 | 0.11 | 0.10 | 0.07 | 0.04 | 0.01 | 0 | 0 |

Table 12: SynBench-Score comparisons on the finetuning procedure in pretraining on synthetic data with heptadiagonal covariance.

| $a_t = 0.7$ | $\epsilon = 0$ | $\epsilon = 0.1$ | $\epsilon = 0.2$ | $\epsilon = 0.3$ | $\epsilon = 0.4$ | $\epsilon = 0.5$ | $\epsilon = 0.6$ | $\epsilon = 0.7$ | $\epsilon = 0.8$ |
|---|---|---|---|---|---|---|---|---|---|
| ViT-Ti/16 | 0 | 0 | 0 | 0 | 0 | 0 | 0 | 0 | 0 |
| ViT-B/16 | 0.18 | 0.22 | 0.24 | 0.23 | 0.20 | 0.15 | 0.10 | 0.05 | 0.01 |
| ViT-L/16 | 0.18 | 0.24 | 0.28 | 0.29 | 0.28 | 0.27 | 0.23 | 0.18 | 0.12 |

Table 13: SynBench-Score comparisons on the model sizes on synthetic data with heptadiagonal covariance.

## A.12 REJECTION MECHANISM

SynBench is a task-agnostic benchmark and it is designed to be used to test pretrained models without the prior knowledge of the downstream task (e.g. model auditing etc). In the case when we do know

| $a_t = 0.7$ | $\epsilon = 0$ | $\epsilon = 0.1$ | $\epsilon = 0.2$ | $\epsilon = 0.3$ | $\epsilon = 0.4$ | $\epsilon = 0.5$ | $\epsilon = 0.6$ | $\epsilon = 0.7$ | $\epsilon = 0.8$ |
|---|---|---|---|---|---|---|---|---|---|
| ViT-S/16-DINO | 0.47 | 0.47 | 0.46 | 0.44 | 0.39 | 0.31 | 0.23 | 0.13 | 0.03 |
| ViT-B/16-DINO | 0.42 | 0.50 | 0.52 | 0.52 | 0.51 | 0.48 | 0.45 | 0.40 | 0.35 |
| ViT-S/8-DINO | 0.36 | 0.38 | 0.38 | 0.38 | 0.36 | 0.33 | 0.30 | 0.26 | 0.20 |
| ViT-B/8-DINO | 0.42 | 0.52 | 0.55 | 0.53 | 0.50 | 0.45 | 0.40 | 0.33 | 0.28 |
| Res50-SimCLRv2 | 0.24 | 0.53 | 0.47 | 0.38 | 0.36 | 0.34 | 0.33 | 0.32 | 0.31 |
| Res101-SimCLRv2 | 0.30 | 0.47 | 0.37 | 0.34 | 0.32 | 0.31 | 0.30 | 0.29 | 0.29 |

Table 14: SynBench-Scores of self-supervised pretrained representations on synthetic data with heptadiagonal covariance.

some knowledge of the tasks, e.g. pixel dependencies, one can use the knowledge to fine-tune the GMM SynBench uses. However, in the case when we know exactly which downstream task will we do and the downstream datasets are accessible and representative,, the best practice is to direclty to apply linear probing. If we are to come up with a rejection mechanism, then one can potentially use goodness-of-fit tests to verify the null hypothesis that the downstream data of interest are generated from a Normal distribution. If the data follow Normal distribution, the Mahalanobis distances should follow a Chi-Squared distribution with degrees of freedom equal to the number of features. Then since the CDF for the appropriate degrees of freedom gives the probability of having obtained a value less extreme than this point, subtracting the CDF value from 1 gives the p-value. We conduct the experiment for CIFAR10, SVHN, and TinyImageNet, and report the p-values in Table R1. Because these p-values are high, we can't reject this hypothesis. But if the p-value is below a threshold, one can reject this hypothesis.

| Dataset | Gaussian-I | Gaussian-H |
|---|---|---|
| CIFAR10 | 0.37 | 0.65 |
| SVHN | 0.58 | 0.83 |
| TinyImageNet | 0.31 | 0.51 |

Table 15: The p-values in the hypothesis testing for Gaussian-I and Gaussian-H distributions.

## A.13 SYNTHETIC DATA GENERATION AND SEPERABILITY

The synthetic data can be generated pixel by pixel if the covariance matrix is a diagonal matrix. In the case when the covariance is not a diagonal (like Section 4.4), we need to draw the whole image (or each channel as in Section 4.4) at once from the multivariate normal with generic covariance matrix.

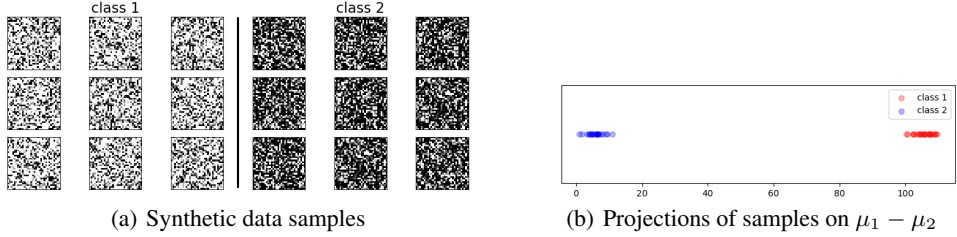

(a) Synthetic data samples        (b) Projections of samples on $\mu_1 - \mu_2$

Figure 6: 18 synthetic data samples and their projections on the direction $\mu_1 - \mu_2$.

We include 18 synthetic data samples in Figure 6(a), showing 9 samples for each of the two classes. These examples are drawn from class-conditional Gaussians with scale $s = 25$ (cf. Section 3.3) and of size $32 \times 32$. Class-1 samples are on the left, and Class-2 samples are on the right. We can see that Class-1 samples are generally brighter than Class-2 samples. This is because Class-1 samples are drawn from the Gaussian with larger mean in the magnitude.

Furthermore, we demonstrate the seperability of two class samples by projecting samples down along the direction of two Gaussian mean difference, in order to showcase their hidden discriminate pattens. That is, for vectorized sample $x$, Gaussian mean $\mu_1$ and $\mu_2$, we do the calculation $x^T(\mu_1 - \mu_2)$ and

plot them on a line in Figure 6(b). From the plot, one can see that the samples from the two classes can be separated easily.

## A.14 MORE FOUNDATION MODELS

We added more network architectures from the pretrained PyTorch Image Models[3]. From Table 16 and 17, we can see that the model performance improves when the size of swin transformer/clip transformer grows, e.g. swin-base has lower SynBench-Score compared with swin-large (0.25 vs 0.27), vit-base has lower SynBench-Score compared with vit-large (0.18 vs 0.25). Also, swin transformers benefit from pretraining on a larger dataset (e.g. "swinv2_base_window12to16_192to256_22kft1k" is pretrained on ImageNet21k before finetuned on ImageNet1k, while "swinv2_base_window16_256" is directly trained on ImageNet1k). Our SynBench-Score also well correlates with the accuracy on ImageNet-1K (fintuning accuracy for Swin transfermers and linear probing accuracy for CLIP image towers). Additional randomly picked models includes EVA and CoCa models, which we list in Table 18.

| Models | SynBench-Score ($\epsilon = 0$) | ImageNet top-1 fine-tuned acc. |
|---|---|---|
| swinv2_base_window16_256 | 0.21 | 84.5 |
| swinv2_base_window12to16_192to256_22kft1k | 0.25 | 86.4 |
| swinv2_large_window12to16_192to256_22kft1k | 0.27 | 87.3 |

Table 16: The SynBench-Score of Swin transformers. ImageNet top-1 accuracy is quoted from Liu et al. (2021).

| Models | SynBench-Score ($\epsilon = 0$) | ImageNet top-1 linear probing acc. |
|---|---|---|
| vit_base_patch16_clip_224.openai | 0.18 | 80.2 |
| vit_large_patch14_clip_224.openai | 0.25 | 83.9 |

Table 17: The SynBench-Score of CLIP image towers. ImageNet top-1 accuracy is quoted from Radford et al. (2021).

| Models | SynBench-Score ($\epsilon = 0$) | ImageNet zero-shot acc. |
|---|---|---|
| eva02_base_patch16_clip_224.merged2b | 0.110 | 74.7 |
| CoCa ViT-B-32 | 0.436 | 82.6 |

Table 18: The SynBench-Score of misc models.

---

[3]https://github.com/huggingface/pytorch-image-models.

A.15   PRETRAIN DATA VERSUS SYNTHETIC DATA

Conducting evaluation with pre-train data can be infeasible/inappropriate due to three reasons. First of all, with the increasing use of self-supervision during the pretraining, the pre-train data can be unlabeled. Secondly, even in the case when the application scenerio is model training and the pre-train data is labeled, the evaluation scores based on the pre-train data can be inconclusive if the evaluation data are biased or under-representative (e.g. pretrained models tend to overfit to the pre-train data). Lastly, from the perspective of the model auditing, the data used for model pretraining can simply be private or inaccessible (e.g., Web-scale raw data).

In these scenarios, one can use SynBench to generate diverse pseudo tasks and non-private synthetic data for conducting comprehensive evaluation of a pre-trained model. By comparing to an idealized data distribution and the corresponding theoretically-optimial reference, SynBench-Score (as illustrated in Figure 1) can quantify the quality of representations, in the sense that the area under the curve (AUC) ratio closer to 1 means better representations.

A.16   LIMITATIONS

**Linear probing.**   SynBench analysis focuses on linear probing performance, which is a popular, low-complexity evaluation protocal widely used in the community (Chen et al., 2020b; He et al., 2020), especially for large neural networks (foundation models). Other assessment tools of pretrained models, such as LogME (You et al., 2021), is also evaluated by the correlation coefficient between their metric and linear probing accuracy. For tasks other than classification, we do observe in some literature that SynBench-Score might still be informative, e.g. ViT-L/16 is reportedly performing worse than ViT-B/16 with MLA decoder in a food segmentation task from Wu et al. (2021), DINO ViT-B performs better than DINO ViT-S in DAVIS 2017 Video object segmentation, and DINO ViT-S/16 performs better than DINO ViT-S/8 according to Jaccard similarity on PASCAL VOC12 dataset from Caron et al. (2021). For fine-tuned pretrain representations, ViT-L/16 loses to ViT-B/16 on finetuned medical tasks with, e.g., X-ray images (Okolo et al., 2022, Table 4-8), and magnetic resonance imaging (Tummala et al., 2022, Table 2-3). Although we are unable to fully justify the relationship between SynBench-Score and non-classification tasks, we believe that if non-classification tasks such as object detection/regression can be translated into classification tasks, SynBench can be extended to those tasks.

**GMM.**   "Can we trust the data representations from a pretrained image model, if it fails to have reasonable performance on simple synthetic datasets?" This is the motivation for our work. When designing the task-agnostic and data-free framework, we want to narrow our scope for a more "well-posed" problem, by using an idealized data distribution with tractable seperability, lifting the need for real-life data. This enables interesting application scenerio such as model auditing, selection, training, and alignment. Therefore, ideologically, SynBench allows any idealized data distribution, provided that the optimal performance (e.g. accuracy-robustness as in our case) can be characterized. At the current stage, the practicality of SynBench owes to the idealized distribution, GMM, whose optimal robust Bayes classifier is known. Our use of GMM for our synthetic input data and modelling their representations is supported by its capablity of modeling the statistics of natural images (Zoran & Weiss, 2012) and prior arts that model hidden layers as GMMs (Zong et al., 2018; Tüske et al., 2015). In our paper, we also try to exemplify how to use more complex covariance to better capture the downstream tasks for specific tasks (Sec. 4.4).

A.17   INTUITIONS ON HOW SYNBENCH PREDICT CLASSIFICATION PERFORMANCE ACROSS A BROAD RANGE OF TASKS

Think of how representation learning research typically evaluate a model for transfer learning - by running tests on broad range of downstream tasks. And the reason behind this is to see how the model behaves in different scenerios. To theorize things, we believe the general behavior of a pretrained representation is measured by how it perform on tasks of different difficulty levels. That is why we think a fundamental part of our design is to simulate tasks of different difficulty levels. One difference between SynBench and a traditional probing test is that, for example, we are using the

classification problem of two highly overlapped Gassuain, instead of classifying ImageNet21k. We hope this clarification builds enough intuition to understand the following:

1. We vary $s$ in equation 2 from 0.1 to 5 in increments of 0.1, which correspond to optimal accuracy (groundtruth difficulty) ranging from 55% to 100% and 50 difficulty levels. If we refer to Figure 7, we see each of the red points correspond to one of our similated trials with difficulty levels (x-axis).

2. Baseline methods are task/data dependant, which means they are somewhat bound to tasks of that similar difficulty levels. If we refer to Figure 7, it could be the single purple point with fixed level of difficulty.

3. If we include certain knowledge of possible downstream data properties, say locality of pixel dependencies, then the prediction will indeed be more accurate (see our section 4.4).

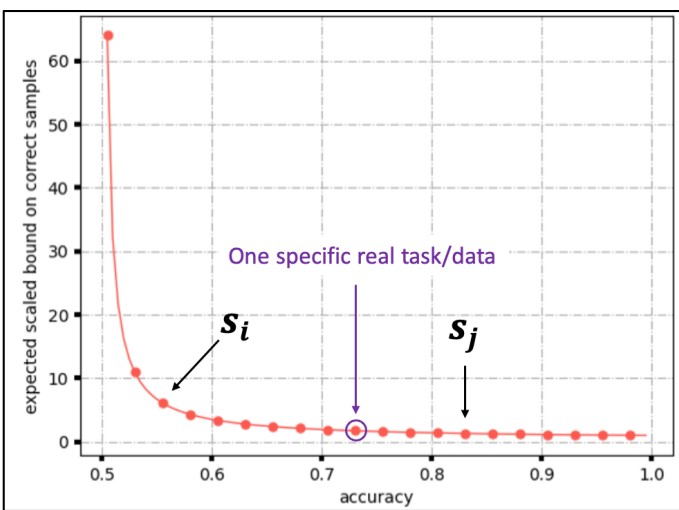

Figure 7: Illustrations of the difference between SynBench synthetic data difficulty coverage and a specific real task/data.

## A.18 CORRELATION BREAKDOWNS AND ROBUSTNESS TO OOD AND CHALLENGING TASKS

As SynthBench score is not dependant on task, we gave the SynthBench score of each model in Table 7. We calculate how SynthBench score correlates with downstream performance per data set in the following Table 19.

| Datasets | Food101 | CIFAR10 | CIFAR100 | birdsnap | SUN397 | StanfordCars | Aircraft |
|---|---|---|---|---|---|---|---|
| FID to ImageNet21k | 100.81 | 115.47 | 96.22 | 102.39 | 54.78 | 154.81 | 206.47 |
| correlation | **0.01** | -0.30 | -0.50 | -0.33 | -0.32 | **0.90** | 0.87 |
| Val loss | -0.31 | 0.07 | 0.24 | 0.03 | 0.03 | -0.82 | -0.70 |
| MDL | -0.18 | **0.19** | **0.37** | **0.17** | **0.16** | -0.84 | -0.77 |
| LogME | -0.48 | -0.70 | -0.83 | -0.74 | -0.74 | 0.85 | **0.95** |
| SFDA | -0.41 | -0.66 | -0.77 | -0.67 | -0.69 | 0.88 | **0.95** |
| Datasets | VOC2007 | DTD | Pets | Caltech101 | Flowers | MNIST | FER2013 |
| FID to ImageNet21k | 52.30 | 98.37 | 104.15 | 53.51 | 112.64 | 301.28 | 175.75 |
| correlation | **0.64** | 0.86 | **0.40** | **0.09** | -0.64 | 0.56 | **0.81** |
| Val loss | -0.80 | -0.66 | -0.63 | 0.02 | 0.37 | -0.33 | -0.85 |
| MDL | -0.76 | -0.75 | -0.54 | -0.01 | **0.49** | -0.41 | -0.82 |
| LogME | 0.22 | **0.98** | -0.13 | -0.01 | -0.92 | **0.85** | 0.55 |
| SFDA | 0.24 | 0.96 | -0.07 | -0.07 | -0.87 | 0.84 | 0.60 |
| Datasets | STL10 | EuroSAT | RESISC45 | GTSRB | KITTI | Country211 | PCAM |
| FID to ImageNet21k | 71.19 | 142.62 | 104.80 | 156.81 | 163.92 | 36.72 | 235.63 |
| correlation | -0.40 | 0.77 | 0.91 | 0.59 | 0.40 | **0.96** | **0.90** |
| Val loss | 0.11 | -0.54 | -0.76 | -0.34 | -0.14 | -0.96 | -0.99 |
| MDL | **0.23** | -0.64 | -0.82 | -0.43 | -0.25 | -0.97 | -0.96 |
| LogME | -0.80 | **0.97** | **0.96** | **0.85** | **0.81** | 0.69 | 0.59 |
| SFDA | -0.75 | 0.93 | **0.96** | 0.82 | 0.77 | 0.70 | 0.64 |
| Datasets | UCF101 | Kinetics700 | CLEVR | HatefulMemes | SST | ImageNet | AVG acc. |
| FID to ImageNet21k | 79.40 | time out | 194.64 | 86.64 | 368.13 | 17.78 | |
| correlation | **0.81** | **0.64** | 0.72 | -0.59 | 0.35 | **0.30** | **0.92** |
| Val loss | -0.93 | -0.82 | -0.48 | 0.34 | -0.22 | -0.56 | -0.92 |
| MDL | -0.87 | -0.74 | -0.59 | **0.47** | -0.32 | -0.45 | -0.91 |
| LogME | 0.45 | 0.17 | **0.97** | -0.88 | **0.41** | -0.22 | 0.72 |
| SFDA | 0.51 | 0.24 | 0.94 | -0.83 | 0.34 | -0.15 | 0.77 |

Table 19: The correlation between SynBench-score and individual downstream task, and the Frechet Inception Distance (FID) scores from ImageNet21k to individual downstream task.

**Subset of OOD tasks**   We further analyze SynBench score's correlation to the subset of OOD tasks. In the following Table 19, we computed the Frechet Inception Distance (FID) scores from ImageNet21k to the downstream tasks, and used them as the indicator of how OOD are the tasks. We then computed SynBench-score correlation with tasks that have FID scores larger than a threshold {50,100,150,200}. We do want to note that not all models in our analysis are pretrained with ImageNet21k; however, since ImageNet21k has become a go-to pretraining dataset, we assume samples therein are in-distribution.
From Table 20, we see that if we don't apply filter on FID (or equivelantly let threshold be 0), the initial correlation was 0.92. As we gradually increase the threshold to 50, 100, 150, and even 200, the correlation stays above 0.8, indeed suggesting SynBench's robustness to OOD tasks.

| FID | > 0 (all tasks) | > 50 | >100 | >150 | > 200 |
|---|---|---|---|---|---|
| SynBench Correlation | 0.92 | 0.93 | 0.93 | 0.82 | 0.92 |

Table 20: The correlation between SynBench-score and the average accuracy of FID-thresholded downstream tasks.

**Subset of more challenging tasks**   We futher analyze SynBench score's correlation to the subset of more challenging tasks. When we check how SynBench can serve as a performance metric of pretrained models, we used the average accuracy of 27 downstream tasks as the proxy of the general performance. Among the 27 tasks, there are indeed datasets that are large and complex, inclduing

| Large/complex datasets | datasets w/ #classes>100 | video datasets (UCF101 and Kinetics 700) | visual reasoning/QA dataset | dataset average |
|---|---|---|---|---|
| SynBench | **0.56** | **0.72** | 0.72 | **0.80** |
| Val loss | -0.75 | -0.88 | -0.48 | -0.91 |
| MDL | -0.66 | -0.81 | -0.59 | -0.85 |
| LogME | 0.11 | 0.30 | **0.97** | 0.45 |
| SFDA | 0.19 | 0.36 | 0.94 | 0.51 |

Table 21: The correlation between SynBench-score and subsets of downstream tasks.

ImageNet. In the following Table 21, we highlight 3 subsets of tasks that represent more challenging datsets in different dimensions (number of classes, data types, task types).

1. For datasets that have more than 100 classes (Food101, Birdsnap, SUN397, StanfordCars, Aircraft, Caltech101, Flowers, Country211, UCF101, Kinetics700, ImageNet), SynBench-score correlates with their average performance with correlation of 0.56, compared with the best baseline (SFDA) of 0.19.

2. For video datasets (UCF101 and Kinetics 700), SynBench-score correlates with their acerage performance with correlation of 0.72, compared with the best baseline (SFDA) of 0.36.

3. For the visual reasoning and question-answering dataset, CLEVR,, SynBench-score correlates with its performance with correlation of 0.72, while LogME and SFDA demonstrate even stronger correlation ($> 0.9$).

Overall, SynBench shows robust performance across these break-down groups.

### A.19 PEARSON AND CONFIDENCE INTERVAL

Let $r$ be the Pearson correlation coefficient, $p$ be the number of models. We ran the calculation for confidence intervals and see that the upper and lower confidence interval limits in z-space are $0.5 \ln(\frac{1+r}{1-r}) \pm 1.96 \sqrt{\frac{1}{p-3}} = 1.589 \pm 1.386$. Translating to r-space by $r = \frac{e^{2z}-1}{e^{2z}+1}$ yields the upper limit of 0.995 and the lower limit of 0.203, if the desired confidence level is $95\%$. In the following Table 22, we added four efficient nets' SynBench-scores, together with the average of their reported performance on 27 downstream tasks in Radford et al. (2021), Table 10. We ran the same calculation for the Pearson correlation coefficient $r = 0.87$ and $p = 9$ to obtain the confidence interval of $[0.488, 0.972]$ which suggest at least moderate correlation upto strong corelation.

| | ViT-B/16 | ViT-L/16 | ViT-B/32 | Resnet50-SimCLRv2 | Resnet101-SimCLRv2 | EfficientNet b0 | EfficientNet b1 | EfficientNet b2 | EfficientNet b3 | Pearson correlation |
|---|---|---|---|---|---|---|---|---|---|---|
| Accuracy (%) | 74.3 | 75.5 | 72.6 | 75.4 | 75.4 | 72.5 | 72.6 | 73.1 | 73.9 | 1.0 |
| SynBench (n=2048) | 0.33 | 0.26 | 0.0 | 0.66 | 0.60 | 0.02 | 0.04 | 0 | 0 | 0.85 |
| SynBench (n=8291) | 0.52 | 0.49 | 0.01 | 0.69 | 0.84 | 0.13 | 0.13 | 0.09 | 0.03 | 0.87 |

Table 22: The correlation between SynBench-score and the average accuracy on 27 real-life tasks.

