# OpenReview forum: "SynBench: Evaluating Pretrained Representations for Image Classification using Synthetic Data"
_ICLR.cc/2024/Conference — Submitted to ICLR 2024_

### Official Review · Reviewer_af4U · 2023-10-28

**Soundness:** 1 poor
**Presentation:** 3 good
**Contribution:** 2 fair
**Rating:** 3
**Confidence:** 4

**Summary:**

The paper considers the problem of evaluating the quality of a pretrained model (for classification) without any data from or knowledge of the downstream tasks for which it will be used. The idea is to generate binary classification data from a Gaussian mixture model and evaluate how well the representation separates the two classes relative to a theoretically optimal classifier. Models are evaluated in terms of standard accuracy and $\epsilon$-robust accuracy (in a certain adversarial sense). Depending on the experiment, evaluation metrics are based on CIFAR10, SVHN, TinyImageNet, or the average performance over the 27 benchmark tasks from the CLIP paper.

Generally, this paper was interesting to read and clear, but there are some issues that need to be addressed before it is ready for publication.

**Strengths:**

* The goal of the paper is ambitious. If the paper solved the problem they pose, it would be very impactful.
* The paper considers its data and results from a number of interesting perspectives, speaking to a general concern for thorough evaluation.
* The idea put forth in the paper is interesting and worthy of further exploration.
* The paper is well-edited and clearly written.

**Weaknesses:**

* The paper assumes that the raw input data for each class has a Gaussian distribution. In computer vision, this means assuming that, in pixel space, two visual categories have Gaussian distributions. This is akin to assuming we start with quite a good representation! What's the point of representation learning if that's where we start? Why would you transform your data?
* One of the key claims of the paper is that the "Pearson correlation between SynBench-Scores and the average real-life task accuracy is larger than 0.9". Flipping to Table 7 in the appendix, we learn that these Pearson correlations are based on $n=5$ data points! It is statistically unacceptable to make this claim without reporting confidence intervals. Running through the [standard calculations](http://faculty.washington.edu/gloftus/P317-318/Useful_Information/r_to_z/PearsonrCIs.pdf), it seems to me that the confidence interval for the Pearson correlation of 0.92 would be $[-0.81, 1.0]$. That is, this correlation value is not very meaningful. This seems to be a highly misleading error, but I'm eager to be corrected if I'm off base on this point. In the absence of this claim, there is little evidence that "real world" task performance is related to the scores computed in this paper.
* There are many hyperparameters in this paper and no mention of hyperparameter tuning, e.g "$a_t$ ranging from 0.7 to 0.9" or "$\epsilon$ from 0 to 0.8" or "attack strength 0.2" etc. Can details be provided? If not, the reader should probably assume that the hyperparameters are charitable to the proposed method.
* SimCLRv2 has the highest SynBench-Score in Table 1, but is generally understood to perform more poorly than the other techniques in the table in common evaluation protocols (see e.g the papers for DINO, BYOL, MAE, or other more recent methods). No mention is made of this in the discussion. This seems like a fairly large problem for the proposed method.
* The paper considers CIFAR10 and TinyImageNet as "real-life downstream data" (see Sec. 4.3) - I think many in computer vision would disagree with this characterization.
* The paper cites Zoran and Weiss 2012 to support the claim that natural image statistics can be well-represented by GMMs. However, that work focuses only on small image patches and (as far as I know) makes no claim that entire natural images can be well-represented by GMMs. Isn't this a misleading use of that reference? (Appendix A.1).

**Questions:**

See "Weaknesses" for supporting details on these questions.
1. A few big picture conceptual questions: Why should we even think it's possible to evaluate the quality of a representation for arbitrary classification tasks without any knowledge or data related to downstream tasks for which it will be used? Moreover, why should we think it's possible to boil that down to a single scalar? What does "quality" even mean without reference to downstream tasks?
2. Are the Pearson correlations appropriately reported (see weaknesses)? Absent that, is there strong evidence that the method in the paper produces scores that are predictive of performance on downstream tasks?
3. Why is it not a problem for the proposed method that SimCLR has the highest score in Table 1, despite being generally understood to underperform DINO and supervised pretraining?
4. The Gaussian assumption is basically an assumption that we start with a very good representation - this seems pretty unrealistic. Why would we bother transforming our data with a pretrained model at that point?
5. Is the reference Zoran and Weiss used appropriately in Appendix A.1?
6. How were the hyperparameters in this paper tuned?

---

> ### Author Response · Authors · 2023-11-19
> **We thank the reviewer for the comments and we believe there are many points that we need to clarify. We wish our explanations can eliminate doubts. (1/3)**
>
> ## 1. Why Gaussuian distribution
> We view SynBench as a "necessary" and "minimum" model test. We concur with the reviewer's argument that assuming a Gaussian distribution for raw input data in pixel space implies starting with a significantly structured and "good" representation. This assumption is intentional and serves the specific purpose of establishing a foundational test. If a pretrained representation network cannot preserve the quality of data drawn from such simple, well-defined distribution (indicated by a SynBench-score close to 1), it raises significant concerns about its potential performance in more complex, real-world scenarios. In our test, we do see that the seperability of the representations could degrade by significant margins compared with their reference behavior in the well-defined simple input space.
>
> To draw an analogy for easier understanding, consider a hgh-performing language model. One might test its basic capabilities by checking whether it correctly writes "2" after "1+1=" or classifies "bad" as a negative word. Failure to perform these straightforward tasks with clear expected outcomes would suggest fundamental deficiencies in the model.
>
> Moreover, we also believe that our designed task with conditional Gaussian is feasible. For a feature extractor robust to noises/good at denoising, all class-1 synthetic samples shall be treated as point $\mu$ and all class-2 samples as point $-\mu$, which are then trivial to classify.
>
> ## 2. Pearson and confidence interval
> Let $r$ be the Pearson correlation coefficient, $p$ be the number of models. We ran the calculation based on the reference shared by the reviewer, and see that the upper and lower confidence interval limits in z-space are $0.5\ln(\frac{1+r}{1-r}) \pm 1.96\sqrt{\frac{1}{p-3}}=1.589\pm 1.386$. Translating to r-space by $r=\frac{e^{2z}-1}{e^{2z}+1}$ yields the upper limit of 0.995 and the lower limit of 0.203. That said, we believe the confidence interval for the Pearson correlation of 0.92 is instead $[0.203,0.995]$ if the desired confidence level is $95\%$. We are not sure how the reviewer obtained $[-0.81,1.0]$ and wish the reviewer can let us know to make sure we are on the same page.
>
> In the following Table 4.1, we added four efficient nets' SynBench-scores, together with the average of their reported performance on 27 downstream tasks in [Radford et al., 2021](https://arxiv.org/pdf/2103.00020.pdf), Table 10. We ran the same calculation for the Pearson correlation coefficient $r=0.87$ and $p=9$ to obtain the confidence interval of $[0.488,0.972]$ which suggest at least moderate correlation upto strong corelation.
>
> | | ViT-B/16 | ViT-L/16 | ViT-B/32 | Resnet50-SimCLRv2 | Resnet101-SimCLRv2 | ENet_b0   | ENet_b1   | ENet_b2   | ENet_b3   | Pearson correlation |
> |----------|----------|----------|----------|-------------------|--------------------|------|------|------|------|---------------------|
> | Accuracy (%) | 74.3     | 75.5     | 72.6     | 75.4              | 75.4               | 72.5 | 72.6 | 73.1 | 73.9 | 1.0                 |
> | SynBench (n=2048)  | 0.33     | 0.26     | 0.0      | 0.66              | 0.60               | 0.02 | 0.04 | 0    | 0    | 0.85                |
> | SynBench (n=4096)  | 0.52     | 0.49     | 0.01     | 0.69              | 0.84               | 0.13 | 0.13 | 0.09 | 0.03 | 0.87                |
>
> Table 4.1 The correlation between SynBench-score and the average accuracy on 27 real-life tasks.
>
>
> ## 3. Role of hyperparameters
>
> *Threshold accuracy $a_t$*. $a_t$ is a user-specified threshold accuracy based on their utility requirement. To give a complete picture, we calculate all SynBench-score from 0.7 to 0.9 in table 6 in the paper. By referring to the table, users can see which model might be more suitable for him/her. Throughout our analysis, we used $a_t=0.7$ since in average the reallife tasks accuracy is above 70% (Table 7 in the paper).
>
> *Robust margin $\epsilon$*. Recall that $\epsilon$-robust Bayes optimal classifier give the optimal classifier if the expected perturbation norm is subject to $\epsilon$ budget. When SynBench is used to evaluate clean accuracy without adversarial attacks as in Section 4.2, we let $\epsilon=0$. When we want to perform $\epsilon$-robust linear probing to increase the robustness against adversarial examples as in Section 4.3, we let $\epsilon=\text{argmax}_\epsilon \text{SynBench-Score}$.
>
> *Attack strength*. Throughout our analysis, we let the attack strength be 0.2. This is guided by our wish to distinguish each model's robustness level to adversarial attacks. If the attack strength is too strong, the robust accuracy of all models will decrease to near zero, while if the attack is too weak, no misclassification will occur. We note that this hyperparameter is only for attack evaluation, and it won't affect the computation of SynBench score.

---

> ### Author Response · Authors · 2023-11-19
> **We thank the reviewer for the comments and we believe there are many points that we need to clarify. We wish our explanations can eliminate doubts. (2/3)**
>
> ## 4. SimCLRv2 has higher SynBench-Score than DINO
> We need to note that SynBench is to evaluate specific models, not methodology. As there might have many subtle differences in training parameters/optimizers/datasets, we can only make conclusions for models used in our paper. In our paper, we use SimCLRv2 models from the [simclr official repository](https://github.com/google-research/simclr), specifically, we use SimCLRv2 models with resnet depth 50 and 100, both with width 2X and SK True. For example, if we try to compare with DINO models, we can refer to [DINO official repository](https://github.com/facebookresearch/dino) and see they are generally on par with the specific SimCLRv2 models we considered (see table 4.2). From this example, we don't see these two specific SimCLRv2 models to perform more poorly than DINO, and our paper makes no conclusions for BYOL and MAE. As such, we do not see there is a problem for our proposed method.
>
> | | ViT-S/16-DINO | ViT-S/8-DINO | ViT-B/16-DINO |ViT-B/8-DINO | ResNet50-SimCLRv2 | ResNet101-SimCLRv2 |
> |----------|----------|----------|----------|-------------------|--------------------|------|
> ImageNet acc. | 77.0% | 79.7% | 78.2% | 80.1% | 77.7% | 79.0%|
>
> Table 4.2 The linear probing accuracy on ImageNet1k reported in the original repository.
>
>
> ##  5. CIFAR10 and TinyImageNet as "real-life downstream data" in Section 4.3
> These are popular CV downstream datasets. We have updated our paper accordingly.
>
> ## 6. GMMs' capability of modeling the statistics of natural images
> We believe the claim the reviewer is referring to is "Our use of Gaussian mixtures for analysis is supported by its capability of modeling the statistics of natural images (Zoran \& Weiss, 2012) and prior arts ......". Since our target is pretrained vision models, the capabilities of preceiving contrasts and edges etc are centric (this is also the motivation for convolutional kernels for ResNets, images patches for ViTs etc). From the [reference](https://papers.nips.cc/paper_files/paper/2012/file/e97ee2054defb209c35fe4dc94599061-Paper.pdf), we quote "Simple Gaussian Mixture Models (GMMs) learned from pixels of natural image patches have been recently shown to be surprisingly strong performers in modeling the statistics of natural images" and it is not hard to see that these statistics include contrast, textures at different scales and orientations, and boundaries of objects in the reference. If the reviewer is curious about how images patches connects to whole images, there are some discussions in the literature, [From Learning Models of Natural Image Patches to Whole Image Restoration](https://people.csail.mit.edu/danielzoran/EPLLICCVCameraReady.pdf) and [From Patches to Natural Images via Hierarchical Dirichlet Processes](https://www.michaelchughes.com/papers/JiHughesSudderth_PracticalBNPWorkshop_2016.pdf). We note that another reason for these seminal works to study from image pathes instead is "Due to the high dimensionality of the images captured by modern cameras" (quote from [From Patches to Images: A Nonparametric Generative Model](https://proceedings.mlr.press/v70/ji17a/ji17a.pdf)). Afterall, at the bare minimum we all know GMMs are universal approximators of densities ([Universal Approximation Using Radial-Basis-Function
> Networks](https://direct.mit.edu/neco/article/3/2/246/5580/Universal-Approximation-Using-Radial-Basis), [Networks for approximation and learning](https://ieeexplore.ieee.org/document/58326)), meaning any smooth density can be approximated with any speciﬁc nonzero amount of error by a Gaussian mixture model with enough components.
>
> To our knowledge, we don't know which part is misleading but we are willing to remove or modify the sentence ("Our use of Gaussian mixtures for analysis is supported by its capability of modeling the statistics of natural images (Zoran \& Weiss, 2012) and prior arts ......") following the reviewer's suggestion.

---

> ### Author Response · Authors · 2023-11-19
> **We thank the reviewer for the comments and we believe there are many points that we need to clarify. We wish our explanations can eliminate doubts. (3/3)**
>
> ## 7. How can SynBench predict classification performance across a broad range of tasks
> Think of how representation learning research typically evaluate a model for transfer learning - by running tests on broad range of downstream tasks. And the reason behind this is to see how the model behaves in different scenerios. To theorize things, we believe the general behavior of a pretrained representation is measured by how it perform on tasks of different difficulty levels. That is why we think a fundamental part of our design is to simulate tasks of different difficulty levels. One difference between SynBench and a traditional probing test is that, for example, we are using the classification problem of two highly overlapped Gassuain, instead of classifying ImageNet21k. We hope this clarification builds enough intuition to understand the following:
> 1. We vary $s$ in equation 2 from 0.1 to 5 in increments of 0.1, which correspond to optimal accuracy (groundtruth difficulty) ranging from 55% to 100% and 50 difficulty levels. If we refer to this [figure](https://imgur.com/a/rH819Jm), we see each of the red points correspond to one of our similated trials with difficulty levels (x-axis).
> 2. Baseline methods are task/data dependant, which means they are somewhat bound to tasks of that similar difficulty levels. If we refer to the same [figure](https://imgur.com/a/rH819Jm), it could be the single purple point with fixed level of difficulty.
> 3. If we include certain knowledge of possible downstream data properties, say locality of pixel dependencies, then the prediction will indeed be more accurate (see our section 4.4).

---

> ### Author Response · Authors · 2023-11-21
> **Feedback on Our Rebuttal**
>
> Dear Reviewer af4U,
>
> We would like to thank you again for your review! As the rebuttal/discussion phase is nearing its end, we want to check in to see if we have dispelled all your concerns, or if there are any additional points we can help clarify.
>
> We understand that the discussion period is short, and your support and feedback are very important to us. We are truly grateful for your insights and the time you've dedicated to reviewing our work.

---

> ### Comment · Reviewer_af4U · 2023-11-22
> **Response to authors**
>
> Thank you for your detailed response! Apologies for getting back to you so late.
>
> My fundamental issue remains. The paper is making a rather extraordinary claim: that assessing performance on separating two Gaussians could be predictive of performance on arbitrary downstream classification tasks. I don't see enough strong evidence to support that extraordinary claim. Some specific responses can be found below.
>
> **Regarding the Gaussian distributions**, I tend to agree with other reviewers that it seems like a fairly contrived task, and it is difficult to see why it would be predictive of performance on arbitrary downstream tasks. It seems like the authors more or less agree (with the "contrived" part, anyway) - based on the authors' response, SynBench is supposed to be a low bar that any reasonable representation should clear. But then shouldn't any model that performs decently well on real-world tasks excel on SynBench? If so, what are we to make of e.g. Table 4.1 in the rebuttal, where the EfficientNet backbones have awful SynBench scores but good real-world accuracies?
>
> **Regarding the confidence intervals**, you are correct that I messed up the calculation and I agree with your numbers. However, I think my point stands. A key claim of the paper - "The Pearson correlation coefficient between SynBench-Scores and the average real-life task accuracy is larger than 0.9" - is pretty misleading if the $95\%$ confidence interval is $[0.203, 0.995]$. Moreover, with such a wide confidence interval, this does not seem to be strong evidence that the proposed approach works much better than other approaches. Based on the confidence intervals we're discussing, it seems like $n=5$ is likely too small to draw strong conclusions about Pearson's $r$. While the additional results in Table 4.1 are interesting, there is no comparison to other methods. The scores also vary wildly between backbones that have similar real-task performance, which raises more questions that it answers.
>
> **Regarding GMMs and natural images**, you are correct that I'm referring to the claim that the paper's claim that the "use of Gaussian mixtures for analysis is supported by its capability of modeling the statistics of natural images". Since this paper uses Gaussian mixtures very differently than Zoran & Weiss (i.e. two Gaussians for the whole image vs. tens of Gaussians for modeling patches) this strikes me as a misleading use of that reference. What evidence in Zoran & Weiss supports modeling entire natural images using a mixture of a small number of Gaussians?
>
> **Regarding model ranking**, I didn't realize the SimCLR models were 2x variants. Probably it's best to update the tables to rename e.g. "Resnet50-SimCLRv2" to something like "Resnet50(2x)-SimCLRv2", since "Resnet50" by itself typically just denotes the 1x version. But there are still ranking reversals - why should the ResNet101 variant have a higher SynBench score than the ResNet50 variant, when the 101 has better performance than the 50? It seems like an important point to understand.
>
> At the moment I lean towards keeping my rating as-is - I think the paper has potential, but currently there are too many question marks that have direct consequences for the central claims of the paper.

---

> ### Author Response · Authors · 2023-11-23
> **We thank the reviewer for getting back to us. (1/2)**
>
> We thank the reviewer for getting back to us. We are also glad that we still have a little time to run the additional results the reviewer wants to see and share more of our thoughts! Your comments and discussions are intrumental to our work.
>
> Before we provide our detailed responses, we would like to use this opportunity to reiterate the overarching goal and contributions of this paper. Our primary objective is to introduce a new task-agnostic framework for evaluating the robustness-accuracy of pretrained representations and to inspire other task-agnostic benchmarking designs. This paper already encompasses a substantial amount of material (what is the metric, how can we use the metric to inform performance, how can we use the metric to select robust linear probing parameters), and we believe it is prudent to reserve some aspects for future exploration and development, including improving its rank efficiency (though SynBench already performs the best among all compared baselines).
>
> > what are we to make of e.g. Table 4.1 in the rebuttal, especially EfficientNet backbones
>
> We want to draw to the reviewer's attention that EfficientNet backbones perform almost the worst on average (except ViT-B/32), and also on some tasks (e.g. Food101, CIFAR10, CIFAR100, SUN397, Caltech101, Flowers, FER2013, Country211, UCF101, Kinetics700). If we check their SynBench scores, we see, when $n=8192$, SynBench scores of EfficientNets are around 0.1 while ViT-B/32 is 0.01. By that, we think SynBench indeed informs the general performance.
>
> > baselinse with additional models
>
> Thank you for confirming with us and we are glad that we are on the same page regarding the calculation. In the following table, we have added the baseline results to make Table 4.1 more complete and have also included the equivalent figure of Figure 4 of the paper [here](https://imgur.com/a/I6M2qz9).
>
> |n| Name | ViT-B/16 | ViT-L/16 | ViT-B/32 | Resnet50(2x)-SimCLRv2 | Resnet101(2x)-SimCLRv2 | ENet_b0   | ENet_b1   | ENet_b2   | ENet_b3   | Pearson correlation |
> |----------|----------|----------|----------|----------|-------------------|--------------------|------|------|------|------|---------------------|
> || Accuracy (%) | 74.3     | 75.5     | 72.6     | 75.4              | 75.4               | 72.5 | 72.6 | 73.1 | 73.9 | 1.0                 |
> |n=2048| SynBench | 0.33     | 0.26     | 0.0      | 0.66              | 0.60               | 0.02 | 0.04 | 0    | 0    | **0.85**                |
> || Val loss | 3.10 | 4.12 | 4.10 | 1.31 | 0.98 | 4.66 | 3.56 | 6.82 | 3.88 | -0.63
> || MDL | 6820.76 | 8094.06 | 8198.55 | 5881.34 | 2882.36 | 8950.38 | 7654.88 | 15816.05 | 8138.87 | -0.53
> ||LogME|-0.726 | -0.724 | -0.729 | 2.791 | 1.503 | -0.721 | -0.726 | -0.725 | -0.729 | 0.67
> ||SFDA| 0.584 | 0.635 | 0.567 | 0.947 | 0.593 | 0.534 | 0.515 | 0.751 | 0.823 | 0.44
> |n=8192| SynBench   | 0.52     | 0.49     | 0.01     | 0.69              | 0.84               | 0.13 | 0.13 | 0.09 | 0.03 | **0.87**                |
> || Val loss | 0.73 | 1.50 | 2.92 | 0.62 | 0.52 | 4.27 | 2.03 | 4.33 | 2.56 | -0.78
> || MDL |9939.13 | 17672.6 | 23332.98 | 9646.09 | 5443.43 | 32511.61 |  19479.78 | 43202.85 | 25964.38 | -0.69
> ||LogME| -0.710 | -0.707 | -0.727 | -0.599 | -0.622 | -0.714 | -0.719 | -0.721 | -0.725 | 0.71
> ||SFDA| 0.525 | 0.531 | 0.513 | 0.581 | 0.543 | 0.510 | 0.505 | 0.524 | 0.525 | 0.78
> |n=32768| SynBench   | 0.59 | 0.58 | 0.02 | 0.81 | 0.87 | 0.19 | 0.19 | 0.17 | 0.09 | **0.88**
> || Val loss |  0.68 | 0.79 | 3.91 | 0.53 | 0.51 | 1.11 | 0.79 | 2.60 | 1.11 | -0.58
> || MDL |30848.99 | 38718.04 | 107960.49 | 22022.08 | 17166.0 | 56621.37 | 39158.90 | 109706.34 | 56621.37 | -0.67
> ||LogME| -0.686  | -0.687  | -0.725  | -0.580  | -0.608 | -0.713 | -0.719 | -0.715 | -0.718 | 0.79
> ||SFDA|0.517 | 0.518 | 0.505 | 0.545 | 0.534 | 0.505 | 0.504 | 0.508 | 0.508 | 0.84
>
> Table 4.1R The correlation between SynBench-score and the average accuracy on 27 real-life tasks.
>
> > GMMs and natural images
>
> We propose to revise the sentence as
>
> *In an earlier work that discusses the connections between GMMs and natural image modeling, it has been demonstrated that "Simple Gaussian Mixture Models (GMMs) learned from pixels of natural image patches have been recently shown to be surprisingly strong performers in modeling the statistics of natural images". Our adoption of Gaussian mixtures for analysis is inspired by this finding, as the image patches in our synthetic data are also Gaussians. Additionally, the use of Gaussian is further supported by their capability to act as universal approximators of densities as well as by existing literature on Gaussian design.*
>
> We are open to modifying or removing this sentence should the reviewer find it necessary.

---

> > ### Author Response · Authors · 2023-11-23
> > **We thank the reviewer for getting back to us. (2/2)**
> >
> > > model ranking
> >
> > Yes, we will rename the model. It does not appear to us that there is a definite winner between ResNet101 and ResNet50 upon checking the reported accuracy in [Radford et al., 2021](https://arxiv.org/pdf/2103.00020.pdf), Table 10. Notably, ResNet101 underperforms compared to ResNet50 in several tasks, including Cars, Aircraft, MNIST, RESISC45, GTSRB, KITTI, and CLEVR, aside from other tasks where their performance is tied.
> >
> > We also want to draw to reviewer's attention that in other research domains such as training-free nueral architecture search (NAS), the same correlation metric is often used to evaluate the predictability of a training-free NAS metric versus the true accuracy on each candidate network. In that scenario, the rankings from training-free NAS metrics can be imperfect (just like our case), but higher correlation indeed suggests better utility of a training-free NAS metric. We hope the analogy of SynBench scores to training-free NAS metrics, along with the other use cases of SynBench (e.g., inform a better design of linear probing), can convince the reviewer on the overall utility of SynBench.

---

### Official Review · Reviewer_4o5T · 2023-10-31

**Soundness:** 2 fair
**Presentation:** 3 good
**Contribution:** 2 fair
**Rating:** 3
**Confidence:** 3

**Summary:**

The paper introduces a task-agnostic evaluation framework based on synthetic data to estimate how well pretrained representations transfer to downstream tasks, and how robust they are. Concretely, the paper proposes to generate data from a mixture of Gaussians, and to measure how well separable according to mixture correspondence this data is when embedded with a pretrained representation network, compared to how separable the data is in the input space. The paper derives a corresponding theory, proposes a benchmark score and numerically investigates how well this score correlates with the robustness and transferability of different representations to a variety of tasks.

**Strengths:**

Better quality and more efficient evaluation methods are an important area of active research. Model robustness, while having been improved with increased model and data size, is still unsolved. The paper aims to address both these aspects. Further, further since the proposed method relies on synthetic data, it can avoid issues related to privacy and mitigate undesired biases.

**Weaknesses:**

I generally found the paper rather hard to follow. It is often unclear if the authors are targeting adversarial robustness, or how well a representation transfers or both.

I might well have missed a central point of the paper, but I fundamentally doubt that it is possible
1. solely based on Gaussian mixtures with two components,
2. without any knowledge about the target downstream tasks,

to accurately predict classification performance of feature extractors across a broad range of downstream tasks. The baselines such as (Whitney et al., 2020) all rely on measures that are derived from the feature extractor and the downstream task/data, whereas the current method performs the predictions solely based on the feature extractor and synthetic data, while claiming to outperform the baselines.

Another aspect that I found surprising is that the theory does not depend on the properties of the pretrained feature extractor, for example its Lipschitz constant, which is usually the case in similar contexts.

Overall, I cannot recommend acceptance at this point without further clarifications from the authors.

**Questions:**

I could not find any individual results of the proposed method on the 27 tasks from (Radford et al.) (the correlation is plotted in Figure 4). I would be interested to see the SynthBench score per task and model, and how well it correlates with downstream performance per data set.

---

> ### Author Response · Authors · 2023-11-19
> **We thank the reviewer for the comments and we wish our explanations can dispel concerns. (1/3)**
>
> ## 1. SynBench's target
> SynBench's target covers both adversarial robustness and representation seperability (accuracy). To be more precise, we believe a good pretrained representation network should produce representations that entail good seperability, either *with* or *without*, adversarial perturbations. Please see our paper section 1, the second paragraph, for motivations of our scope. SynBench-score as a metric is designed to serve this purpose and perform evaluations. By definition, SynBench-score is the relative AUC of expected bound-threshold accuracy curve (see equation 3). When SynBench is used to evaluate clean accuracy as in Section 4.2, we let $\epsilon=0$, whearas when it is used to suggest robust linear probing parameters $\epsilon$ as in Section 4.3, we let $\epsilon=\text{argmax}_\epsilon \text{SynBench-Score}$.
>
> ## 2. How can SynBench predict classification performance across a broad range of tasks
> Think of how representation learning research typically evaluate a model for transfer learning - by running tests on broad range of downstream tasks. And the reason behind this is to see how the model behaves in different scenerios. To theorize things, we believe the general behavior of a pretrained representation is measured by how it perform on tasks of different difficulty levels. That is why we think a fundamental part of our design is to simulate tasks of different difficulty levels. One difference between SynBench and a traditional probing test is that, for example, we are using the classification problem of two highly overlapped Gassuain, instead of classifying ImageNet21k. We hope this clarification builds enough intuition to understand the following:
> 1. We vary $s$ in equation 2 from 0.1 to 5 in increments of 0.1, which correspond to optimal accuracy (groundtruth difficulty) ranging from 55% to 100% and 50 difficulty levels. If we refer to this [figure](https://imgur.com/a/rH819Jm), we see each of the red points correspond to one of our similated trials with difficulty levels (x-axis).
> 2. Baseline methods are task/data dependant, which means they are somewhat bound to tasks of that similar difficulty levels. If we refer to the same [figure](https://imgur.com/a/rH819Jm), it could be the single purple point with fixed level of difficulty.
> 3. If we include certain knowledge of possible downstream data properties, say locality of pixel dependencies, then the prediction will indeed be more accurate (see our section 4.4).
>
> ## 3. Discussions: dependency on the properties of the pretrained feature extractor
> We thank the reviewer for bringing this up. If possible, can the reviewer please share the literatures mentioned that use properties of the pretrained representations in similar contexts with us? In our theory, the reference optimal behavior is completely independant of the pretrained feature extractor (the denominator in Equation 3), and the evaluation depends on the feature extractor via the numerator. During this process, we do not explicitly use properties such as Lipschitzness. To our knowledge, the connection between Lipschitzness of the representation network and the downstream task accuracy via linear probing is not obvious.

---

> ### Author Response · Authors · 2023-11-19
> **We thank the reviewer for the comments and we wish our explanations can dispel concerns. (2/3)**
>
> ## 4. Correlation breakdowns
> As SynthBench score is not dependant on task, we gave the SynthBench score of each model in Table 7 in the paper. We calculate how SynthBench score correlates with downstream performance per data set in the following Table 3.1.
>
> | Datasets| Food101 | CIFAR10 | CIFAR100 | birdsnap | SUN397 | StanfordCars | Aircraft  |  VOC2007 |  DTD |  Pets | Caltech101 | Flowers | MNIST | FER2013
> | -------- | -------- | -------- | -------- | -------- | -------- | -------- | -------- |-------- | -------- | -------- | -------- | -------- | -------- | --------
> | SynBench correlation | **0.01** | -0.30 | -0.50 | -0.33 | -0.32 | **0.90** | 0.87 | **0.64** | 0.86 | **0.40** | **0.09** | -0.64 | 0.56 | **0.81** |
> |Val loss | -0.31 | 0.07 | 0.24 | 0.03 | 0.03 | -0.82 | -0.70 | -0.80 | -0.66 | -0.63 | 0.02 | 0.37 | -0.33 | -0.85 |
> |MDL| -0.18 | **0.19** | **0.37** | **0.17** | **0.16** | -0.84 | -0.77 | -0.76 | -0.75 | -0.54 | -0.01 | **0.49** | -0.41 | -0.82 |
> |LogME| -0.48 | -0.70 | -0.83 | -0.74 | -0.74 | 0.85 | **0.95** | 0.22 | **0.98** | -0.13 | -0.01 | -0.92 | **0.85** | 0.55 |
> |SFDA| -0.41 | -0.66 | -0.77 | -0.67 | -0.69 | 0.88 | **0.95** | 0.24 | 0.96 | -0.07 | -0.07 | -0.87 | 0.84 | 0.60 |
> | Datasets| STL10  | EuroSAT | RESISC45 | GTSRB | KITTI | Country211 | PCAM | UCF101 | Kinetics700 | CLEVR | HatefulMemes | SST |  ImageNet | AVG acc.|
> | SynBench correlation | -0.40 | 0.77 | 0.91 | 0.59 | 0.40 | **0.96** | **0.90** | **0.81** | **0.64** | 0.72 | -0.59 | 0.35 | **0.30** | **0.92** |
> |Val loss | 0.11 | -0.54 | -0.76 | -0.34 | -0.14 | -0.96 | -0.99 | -0.93 | -0.82 | -0.48 | 0.34 | -0.22 | -0.56 | -0.92 |
> |MDL| **0.23** | -0.64 | -0.82 | -0.43 | -0.25 | -0.97 | -0.96 | -0.87 | -0.74 | -0.59 | **0.47** | -0.32 | -0.45 | -0.91|
> |LogME| -0.80 | **0.97** | **0.96** | **0.85** | **0.81** | 0.69 | 0.59 | 0.45 | 0.17 | **0.97** | -0.88 | **0.41** | -0.22 | 0.72 |
> |SFDA| -0.75 | 0.93 | **0.96** | 0.82 | 0.77 | 0.70 | 0.64 | 0.51 | 0.24 | 0.94 | -0.83 | 0.34 | -0.15 | 0.77 |
>
> Table 3.1 The correlation between SynBench-score and individual downstream task.
>
> ### **Subset of OOD tasks**
> We further analyze SynBench score's correlation to the subset of OOD tasks. In the following Table 3.2, we computed the Frechet Inception Distance (FID) scores from ImageNet21k to the downstream tasks, and used them as the indicator of how OOD are the tasks. We then computed SynBench-score correlation with tasks that have FID scores larger than a threshold {50,100,150,200}. We do want to note that not all models in our analysis are pretrained with ImageNet21k; however, since ImageNet21k has become a go-to pretraining dataset, we assume samples therein are in-distribution.
>
> From table 3.3, we see that if we don't apply filter on FID (or equivelantly let threshold be 0), the initial correlation was 0.92. As we gradually increase the threshold to 50, 100, 150, and even 200, the correlation stays above 0.8, indeed suggesting SynBench's robustness to OOD tasks.
>
> | Datasets| Food101 | CIFAR10 | CIFAR100 | birdsnap | SUN397 | StanfordCars | Aircraft  |  VOC2007 |  DTD |  Pets | Caltech101 | Flowers | MNIST | FER2013 |
> | -------- | -------- | -------- | -------- | -------- | -------- | -------- | -------- |-------- | -------- | -------- | -------- | -------- | -------- | -------- |
> | FID to ImageNet21k| 100.81 | 115.47 | 96.22 | 102.39 | 54.78 | 154.81 | 206.47 | 52.30 | 98.37 | 104.15 | 53.51 | 112.64 | 301.28 |175.75 |
> | Datasets| STL10  | EuroSAT | RESISC45 | GTSRB | KITTI | Country211 | PCAM | UCF101 | Kinetics700 | CLEVR | HatefulMemes | SST |  ImageNet |
> | FID to ImageNet21k| 71.19 | 142.62| 104.80 | 156.81 | 163.92 | 36.72 | 235.63 | 79.40 | time out | 194.64 | 86.64 | 368.13 | 17.78 |
>
> Table 3.2 The Frechet Inception Distance (FID) scores from ImageNet21k to 27 downstream tasks.
>
> | FID| > 0 (all tasks)| > 50 | >100 | >150 | > 200 |
> |-------- |-------- |-------- |-------- |-------- | -------- |
> | SynBench Correlation | 0.92 | 0.93 | 0.93 | 0.82 | 0.92|
>
> Table 3.3 The correlation between SynBench-score and the average accuracy of FID-thresholded downstream tasks.

---

> ### Author Response · Authors · 2023-11-19
> **We thank the reviewer for the comments and we wish our explanations can dispel concerns. (3/3)**
>
> ### **Subset of more challenging tasks**
> We further analyze SynBench score's correlation to the subset of more challenging tasks. When we check how SynBench can serve as a performance metric of pretrained models, we used the average accuracy of 27 downstream tasks as the proxy of the general performance. Among the 27 tasks, there are indeed datasets that are large and complex, inclduing ImageNet. In the following Table 3.4, we highlight 3 subsets of tasks that represent more challenging datsets in different dimensions (number of classes, data types, task types).
> 1. For datasets that have more than 100 classes (Food101, Birdsnap, SUN397, StanfordCars, Aircraft, Caltech101, Flowers, Country211, UCF101, Kinetics700, ImageNet), SynBench-score correlates with their average performance with correlation of 0.56, compared with the best baseline (SFDA) of 0.19.
> 2. For video datasets (UCF101 and Kinetics 700), SynBench-score correlates with their acerage performance with correlation of 0.72, compared with the best baseline (SFDA) of 0.36.
> 3. For the visual reasoning and question-answering dataset, CLEVR,, SynBench-score correlates with its performance with correlation of 0.72, while LogME and SFDA demonstrate even stronger correlation ($>0.9$).
>
> Overall, SynBench shows robust performance across these break-down groups.
>
> |  Large/complex datasets |  datasets w/ #classes>100 |  video datasets (UCF101 and Kinetics 700) |  visual reasoning/QA dataset | dataset average|
> | -------- | -------- | -------- | -------- | -------- |
> | SynBench| **0.56** | **0.72** | 0.72 | **0.80** |
> |Val loss  |-0.75 | -0.88| -0.48 | -0.91 |
> |MDL | -0.66 | -0.81 | -0.59 | -0.85 |
> |LogME | 0.11 | 0.30 |**0.97** | 0.45|
> |SFDA | 0.19 | 0.36|0.94 | 0.51 |
>
> Table 3.4 The correlation between SynBench-score and subsets of downstream tasks.

---

> ### Author Response · Authors · 2023-11-21
> **Feedback on Our Rebuttal**
>
> Dear Reviewer 4o5T,
>
> We would like to thank you again for your review! As the rebuttal/discussion phase is nearing its end, we want to check in to see if we have dispelled all your concerns, or if there are any additional points we can help clarify.
>
> We understand that the discussion period is short, and your support and feedback are very important to us. We are truly grateful for your insights and the time you've dedicated to reviewing our work.

---

> ### Comment · Reviewer_4o5T · 2023-11-21
>
> I thank the authors for their effort in providing a detailed rebuttal. Overall the provided rebuttal material confirmed my original opinion. Here are responses to some of the author's points in the rebuttal.
>
> **2. How can SynBench predict classification performance across a broad range of tasks**
>
> I stand by my original assessment that some information about the downstream tasks is required to make reasonable predictions about the performance of predictors on these tasks. A mixture of two Gaussians in pixel space is an extremely inaccurate model of natural images. As an insightful example the authors can compute the SynBench for the identity function (i.e. directly in pixel space, without any feature extractor) and compare its performance with features extracted with strong image classification models.
>
> **3. Dependency on the properties of the pretrained feature extractor**
>
> The very first paper on adversarial examples (Szegedy et al., 2013) analyzes the stability of the network as a function of the Lipschitz constants of the individual network layers. Many follow-up works rely on similar properties (searching for "Lipschitz" in the 14.7k citing articles of (Szegedy et al., 2013) leads to 1.65k results on Google Scholar).
>
> **4. Correlation breakdowns**
>
> I thank the authors for providing the detailed breakdown. Unfortunately, this confirms that SynBench is not very predictive for the downstream performance across a broad range of tasks. The correlations for the individual data sets are very inconsistent: some are negative, some are positive, some are close to zero.

---

> > ### Author Response · Authors · 2023-11-21
> > **We thank the reviewer for active engagement in the discussion.**
> >
> > We thank the reviewer for active engagement in the discussion.
> >
> > > 2. How can SynBench predict classification performance across a broad range of tasks
> >
> > SynBench is proposed for use in two main scenerio: (1) when we do not have any knowledge about the downsteam task; (2) when we possess some statistical information about the downstream task (e.g., covariance of data).
> > While we see the reviewer's concern pertains to the second scenario, we want to emphasize that our framework works for **arbitrary covariacne matrices that model structures in pixel spaces**. In this regard, we have provided an example in Section 4.4 showing the use of block heptadiagonal matrix to mimic locality. With the modeled covariance, we showed that the SynBench score can provide an even more accurate estimate of downstream performance, imroving the correlation from 0.79 (Gaussian-I) to 0.91 (Gaussian-H) when the synthetic sample size $n=2048$. This result exactly justifies the reviewer's intuition on the utility of SynBench in the second scenario.
> >
> > > 3. Dependency on the properties of the pretrained feature extractor
> >
> > We did not realize the reviewer was referring to robustness literature. Indeed, in quantifying the robustness of end-to-end neural networks, using the network's Lipschitzness is straightforward as it bounds the gradient and can connect variations in the output space back to input space safety regions. In our context, our goal is to assess the accuracy-robustness tradeoff induced by pretrained representations. Therefore, we measure the accuracy and robustness directly on the representations themselves, eliminating the need to propagate back to the input space and hence sparing the need for including the Lipschitz constant of the pretrained extractor.
> >
> > > 4. Correlation breakdowns
> >
> > We respectively disagree with the conclusion drawn. First and foremost, SynBench is designed to assess the **overall performance** of the pretrained representations on a set of basic tasks. If **there exist a metric that highly correlates with the linear probing performance on every single downstream task**, it would imply that **the linear probing performance on every single downstream task also correlates highly with each other**— which is not the case in reality. Therefore, we are seeking a metric that can inform on the potential overall performance. In the following table, we provide the average ranking of correlations with downstream tasks by SynBench and other baselines as a more robust and intuitive measure. It is clear that SynBench is able to give the overall best correlation with each individual downstream.
> >
> > | Correlation | Average ranking
> > | -------- | -------- |
> > | SynBench  | **2.11$\pm$ 0.976**
> > |Val loss |  3.68$\pm$ 1.166
> > |MDL| 3.57$\pm$ 1.613
> > |LogME| 3.00$\pm$ 1.488
> > |SFDA|2.64$\pm$ 1.076

---

### Official Review · Reviewer_97TE · 2023-11-01

**Soundness:** 4 excellent
**Presentation:** 3 good
**Contribution:** 4 excellent
**Rating:** 6
**Confidence:** 2

**Summary:**

The paper proposes SynBench, a method to evaluate the representations of pretrained models using synthetic data. The synthetic dataset is a class-conditional Gaussian in a binary classification setting. The proposed metric measures the accuracy and robustness on this constructed synthetic dataset (proxy task). SynBench-score is then defined as the ratio of area-under-curve between the representations obtained from pretrained models and the reference. The results on various image classification tasks demonstrate that SynBench-Score vastly outperforms baseline methods across wide range of supervised and self-supervised pretrained models. The paper also delves into the potential applications of this metric, discussing scenarios where it could be beneficial.

**Strengths:**

1. The paper provides a very comprehensive set of results with various backbones where SynBench-Score outperforms the baseline methods. The correlation of SynBench-Score is quite high even with limited number of samples.
2. Practically, this can potentially be a very useful metric given that it does not require any real data. The motivation of the paper is well explained and the authors give various scenarios where this metric would be useful.
3. Overall, this is mostly a complete paper with the authors discussing runtime analysis, limitations and the algorithm of SynBench.

**Weaknesses:**

The authors only consider the linear probing paradigm in evaluation of pretrained models. In practice, finetuning is also a common way to use these pretrained models. It is not clear how this metric would perform in the finetuning setup.

**Questions:**

1. What is the number of test samples in the synthetic dataset? I am not sure if I saw this in the paper.
2. In the Algorithm (A.6), what are $z_1$ and $z_2$ in Line 12?
3. It would be interesting to analyse the correlation on a subset of tasks instead of average of 27 tasks. For instance, it may be the case that metric performs better on OOD tasks compared to transfer learning on some datasets.

---

> ### Author Response · Authors · 2023-11-19
> **We thank the reviewer for the positive comments and questions for us. We are happy to clarify the details and give additional explanations.**
>
> ## 1. How SynBench would perform in the finetuning setup
> As discussed in specified in Section 3.3, SynBench is designed to evaluate the pretrained representations parameterized by $\theta$. In the scenerio when finetuning is performed, the parameters of the representation network will be updated, hence resulting in another pretrained representations (a separate model) from the perspective of SynBench. In our paper, we have shown how to use SynBench to compare representation networks before and after finetuning (Section 4.2, Comparing model attributes). In our example, ``ViT-B/16`` and ``ViT-B/16-in21k`` were both pretrained on ImageNet21k with supervision, whereas ``ViT-B/16`` is further finetuned on Imagenet 1k (name convention is adopted from [PyTorch Image Models](https://github.com/huggingface/pytorch-image-models)). The result shows the utility of SynBench in comparing original and finetuned models.
>
> ## 2. The number of test samples in the synthetic dataset
> There are 2048 test samples in the synthetic dataset. We have included this detail to the updated paper.
>
> ## 3. $z_1$ and $z_2$ in Line 12 of Algorithm A.6
> The subscript denotes the class label of the representation. That is, $z_1^{train,i}$ is the $i$-th training sample from class 1, and $z_2^{train,j}$ is the $j$-th training sample from class 2. We have included this detail to the updated paper.
>
> ## 4. How does SynBench inform model performance on OOD tasks vs. others
> We thank the reviewer for raising this question. In the following Table 2.1, we computed the Frechet Inception Distance (FID) scores from ImageNet21k to the downstream tasks, and used them as the indicator of how OOD are the tasks. We then computed SynBench-score correlation with tasks that have FID scores larger than a threshold {50,100,150,200}. We do want to note that not all models in our analysis are pretrained with ImageNet21k; however, since ImageNet21k has become a go-to pretraining dataset, we assume samples therein are in-distribution.
>
> From table 2.2, we see that if we don't apply filter on FID (or equivelantly let threshold be 0), the initial correlation was 0.92. As we gradually increase the threshold to 50, 100, 150, and even 200, the correlation stays above 0.8, indeed suggesting SynBench's robustness to OOD tasks.
>
> | Datasets| Food101 | CIFAR10 | CIFAR100 | birdsnap | SUN397 | StanfordCars | Aircraft  |  VOC2007 |  DTD |  Pets | Caltech101 | Flowers | MNIST | FER2013 |
> | -------- | -------- | -------- | -------- | -------- | -------- | -------- | -------- |-------- | -------- | -------- | -------- | -------- | -------- | -------- |
> | FID to ImageNet21k| 100.81 | 115.47 | 96.22 | 102.39 | 54.78 | 154.81 | 206.47 | 52.30 | 98.37 | 104.15 | 53.51 | 112.64 | 301.28 |175.75 |
> | Datasets| STL10  | EuroSAT | RESISC45 | GTSRB | KITTI | Country211 | PCAM | UCF101 | Kinetics700 | CLEVR | HatefulMemes | SST |  ImageNet |
> | FID to ImageNet21k| 71.19 | 142.62| 104.80 | 156.81 | 163.92 | 36.72 | 235.63 | 79.40 | time out | 194.64 | 86.64 | 368.13 | 17.78 |
>
> Table 2.1 The Frechet Inception Distance (FID) scores from ImageNet21k to 27 downstream tasks.
>
> | FID| > 0 (all tasks)| > 50 | >100 | >150 | > 200 |
> |-------- |-------- |-------- |-------- |-------- | -------- |
> | SynBench Correlation | 0.92 | 0.93 | 0.93 | 0.82 | 0.92|
>
> Table 2.2 The correlation between SynBench-score and the average accuracy of FID-thresholded downstream tasks.

---

> ### Author Response · Authors · 2023-11-21
> **Feedback on Our Rebuttal**
>
> Dear Reviewer 97TE,
>
> We would like to thank you again for your review! As the rebuttal/discussion phase is nearing its end, we want to check in to see if we have dispelled all your concerns, or if there are any additional points we can help clarify.
>
> We understand that the discussion period is short, and your support and feedback are very important to us. We are truly grateful for your insights and the time you've dedicated to reviewing our work.

---

> > ### Comment · Reviewer_97TE · 2023-11-22
> > **Response to Reviewer**
> >
> > I thank the reviewer for the response. Table 2.1 and Table 2.2 discussed in the rebuttal is certainly interesting. Having looked at the valid concerns raised by other reviewers, I'll maintain my score for now.

---

### Official Review · Reviewer_S4rE · 2023-11-01

**Soundness:** 3 good
**Presentation:** 3 good
**Contribution:** 3 good
**Rating:** 6
**Confidence:** 3

**Summary:**

This paper introduces SynBench, a task-agnostic framework designed for the evaluation of pre-trained representations using synthesized data derived from data prior. Notably, SynBench is independent of downstream image classification datasets or tasks. The experimental results demonstrate a strong correlation between SynBench scores and the model's performance as assessed through measures of adversarial robustness and standard accuracy. Additionally, SynBench proves to be helpful for guiding the design and selection of hyperparameters in robust linear probing.

**Strengths:**

- It is interesting to design a proxy task for the quality evaluation of pre-trained representations. This approach offers a fresh perspective on assessing the quality of representations without relying on downstream datasets.

- Experiments show the effectiveness of SynBench-Scores in indicating real-life task accuracy.

- The paper demonstrates a high level of quality in its methodology and experimental design.

**Weaknesses:**

- Robustness to Deviating Distributions

  It would be valuable to assess SynBench's robustness when facing uncommon real-world data distributions. For instance, applying SynBench to datasets like DomainNet, which contains diverse and domain-shifted data, can provide insights into its adaptability to varying data sources and distributions. Demonstrating SynBench's effectiveness under such conditions would strengthen its applicability and reliability.


- Scalability to Tasks with More Classes

   The paper primarily uses datasets with limited categories for experiments. It's important to explore how SynBench performs on tasks with a more extensive range of classes, such as ImageNet.  It would help understand the framework's scalability and whether it maintains its effectiveness when applied to larger and more complex datasets. This expansion of experiments can provide a clearer picture of SynBench's utility across diverse tasks.

**Questions:**

see Weaknesses

---

> ### Author Response · Authors · 2023-11-19
> **We thank the reviewer for the positive comments and questions for us. We hope our additional analysis completes the picture.**
>
> ## 1. How does SynBench inform model performance on OOD tasks vs. others
> We agree that showing SynBench's robustness to OOD tasks will strengthen the applicability and reliability. In the following Table 1.1, we computed the Frechet Inception Distance (FID) scores from ImageNet21k to the downstream tasks, and used them as the indicator of how OOD are the tasks. We then computed SynBench-score correlation with tasks that have FID scores larger than a threshold {50,100,150,200}. We do want to note that not all models in our analysis are pretrained with ImageNet21k; however, since ImageNet21k has become a go-to pretraining dataset, we assume samples therein are in-distribution.
>
> From table 1.2, we see that if we don't apply filter on FID (or equivelantly let threshold be 0), the initial correlation was 0.92. As we gradually increase the threshold to 50, 100, 150, and even 200, the correlation stays above 0.8, indeed suggesting SynBench's robustness to OOD tasks.
>
> | Datasets| Food101 | CIFAR10 | CIFAR100 | birdsnap | SUN397 | StanfordCars | Aircraft  |  VOC2007 |  DTD |  Pets | Caltech101 | Flowers | MNIST | FER2013 |
> | -------- | -------- | -------- | -------- | -------- | -------- | -------- | -------- |-------- | -------- | -------- | -------- | -------- | -------- | -------- |
> | FID to ImageNet21k| 100.81 | 115.47 | 96.22 | 102.39 | 54.78 | 154.81 | 206.47 | 52.30 | 98.37 | 104.15 | 53.51 | 112.64 | 301.28 |175.75 |
> | Datasets| STL10  | EuroSAT | RESISC45 | GTSRB | KITTI | Country211 | PCAM | UCF101 | Kinetics700 | CLEVR | HatefulMemes | SST |  ImageNet |
> | FID to ImageNet21k| 71.19 | 142.62| 104.80 | 156.81 | 163.92 | 36.72 | 235.63 | 79.40 | time out | 194.64 | 86.64 | 368.13 | 17.78 |
>
> Table 1.1 The Frechet Inception Distance (FID) scores from ImageNet21k to 27 downstream tasks.
>
> | FID| > 0 (all tasks)| > 50 | >100 | >150 | > 200 |
> |-------- |-------- |-------- |-------- |-------- | -------- |
> | SynBench Correlation | 0.92 | 0.93 | 0.93 | 0.82 | 0.92|
>
> Table 1.2 The correlation between SynBench-score and the average accuracy of FID-thresholded downstream tasks.
>
> ## 2. Whether SynBench maintains its effectiveness when applied to larger and more complex datasets
> When we check how SynBench can serve as a performance metric of pretrained models, we used the average accuracy of 27 downstream tasks as the proxy of the general performance. Among the 27 tasks, there are indeed datasets that are large and complex, inclduing ImageNet. In the following Table 1.3, we highlight 3 subsets of tasks that represent more challenging datsets in different dimensions (number of classes, data types, task types).
> 1. For datasets that have more than 100 classes (Food101, Birdsnap, SUN397, StanfordCars, Aircraft, Caltech101, Flowers, Country211, UCF101, Kinetics700, ImageNet), SynBench-score correlates with their average performance with correlation of 0.56, compared with the best baseline (SFDA) of 0.19.
> 2. For video datasets (UCF101 and Kinetics 700), SynBench-score correlates with their acerage performance with correlation of 0.72, compared with the best baseline (SFDA) of 0.36.
> 3. For the visual reasoning and question-answering dataset, CLEVR,, SynBench-score correlates with its performance with correlation of 0.72, while LogME and SFDA demonstrate even stronger correlation ($>0.9$).
>
> Overall, SynBench shows robust performance across these break-down groups.
>
> |  Large/complex datasets |  datasets w/ #classes>100 |  video datasets (UCF101 and Kinetics 700) |  visual reasoning/QA dataset | dataset average|
> | -------- | -------- | -------- | -------- | -------- |
> | SynBench| **0.56** | **0.72** | 0.72 | **0.80** |
> |Val loss  |-0.75 | -0.88| -0.48 | -0.91 |
> |MDL | -0.66 | -0.81 | -0.59 | -0.85 |
> |LogME | 0.11 | 0.30 |**0.97** | 0.45|
> |SFDA | 0.19 | 0.36|0.94 | 0.51 |
>
> Table 1.3 The correlation between SynBench-score and subsets of downstream tasks.

---

> > ### Comment · Reviewer_S4rE · 2023-11-23
> >
> > After checking the rebuttals, most of my concerns are solved. I recommend including the additional experiments and discussions during the rebuttal in the revised paper.

---

> ### Author Response · Authors · 2023-11-21
> **Feedback on Our Rebuttal**
>
> Dear Reviewer S4rE,
>
> We would like to thank you again for your review! As the rebuttal/discussion phase is nearing its end, we want to check in to see if we have dispelled all your concerns, or if there are any additional points we can help clarify.
>
> We understand that the discussion period is short, and your support and feedback are very important to us. We are truly grateful for your insights and the time you've dedicated to reviewing our work.

---

### Meta-Review · Area_Chair_WSpG · 2023-12-12

**Metareview:**

**Summary**: The authors propose SynBench, a task-agnostic framework for evaluating the effectiveness and robustness of pre-trained representations in transferring to downstream image classification tasks using synthetic data. It utilizes a class-conditional Gaussian synthetic dataset for binary classification as a proxy task. The SynBench score measures the discrepancy in classifier separability between the input space and the embedding space of a pre-trained representation model. Experimental results show an interesting correlation between SynBench scores and model performance in terms of standard accuracy and adversarial robustness. The method demonstrates efficacy across a range of supervised and self-supervised pretrained models, outperforming baseline methods. In addition, a corresponding theory is derived to support the claim of the paper.

**Strengths And Weaknesses**: From the reviewers’ perspectives, the strengths of the paper are clear: an important problem with an ambitious goal, high potential impact, interesting idea, and extensive evaluation.

On the other hand, there are several concerns raised by the reviewers.

(i) Unconvincing claims. The reviewers expressed skepticism about the main claim in this paper, which involves predicting the performance of pre-trained representations on arbitrary downstream datasets or tasks solely based on binary classification of synthetic data generated from a mixture of two Gaussians. They noted that prior work in this domain typically relies on specific information about the downstream task or data, making the presented claim seem somewhat extraordinary without sufficient justification. Additionally, the reviewers highlighted a lack of supporting evidence for modeling natural images using a mixture of a small number of Gaussians, attributing this to notable setting differences between this paper and prior work. They specifically pointed out imprecisions in the original submission's statement.

(ii) Rigorousness of the empirical evaluation. During the initial assessment, the reviewers raised concerns about the lack of evaluation on out-of-distribution and larger-scale datasets, the absence of reported results for individual tasks, and missing an in-depth discussion on the confidence interval for the Pearson correlation. During the rebuttal phase, the authors conducted additional experiments to address the initial concerns and provided breakdown results. However, these results introduced new concerns for the reviewers. They questioned the solidity of the conclusion based on the confidence intervals and noted inconsistencies in the correlations of the SynthBench score for individual datasets, with some being negative, some positive, and some close to zero.

At the conclusion of the discussion phase, there is no champion of the paper. Two out of four reviewers kept their rejection ratings. The other two reviewers were slightly positive, but emphasized the need for further improvements to meet the acceptance standards. The area chairs found that the weaknesses raised by the reviewers are valid and weighted the paper's weaknesses over the current strengths. Therefore, the area chairs cannot recommend the acceptance of the current paper. But the area chairs believe the paper could be significantly improved by (i) revisiting several statements to enhance clarity and precision, including explicit clarification that SynBench is designed to evaluate the overall performance of pre-trained representations across a set of datasets/tasks rather than individual ones, and addressing the modeling of natural images using a mixture of Gaussians; (ii) offering a more intuitive explanation and in-depth justification of the main claim; and (iii) incorporating additional results obtained during the rebuttal period, expanding on them, and conducting a rigorous analysis.

**Justification For Why Not Higher Score:**

There is no champion of the paper. The area chairs found that the weaknesses raised by the reviewers are valid and weighted the paper's weaknesses over the current strengths. Therefore, the area chairs cannot recommend the acceptance of the current paper. But the area chairs believe the paper will be significantly improved by (i) revisiting several statements to enhance clarity and precision, including explicit clarification that SynBench is designed to evaluate the overall performance of pre-trained representations across a set of datasets/tasks rather than individual ones (this may potentially reduce the method's practical significance, as pre-trained representations are often selected for specific tasks, as well as raising concerns about the selection bias within the set of evaluation datasets), and addressing the modeling of natural images using a mixture of Gaussians; (ii) offering a more intuitive explanation and in-depth justification of the main claim; and (iii) incorporating additional results obtained during the rebuttal period, expanding on them, and conducting a rigorous analysis.

**Justification For Why Not Lower Score:**

N/A

---

### Decision · Program_Chairs · 2024-01-16

Reject